# SAMRS: Scaling-up Remote Sensing Segmentation Dataset with Segment Anything Model

**Di Wang**[1][*] **Jing Zhang**[2][†]**, Bo Du**[1][‡]**, Minqiang Xu**[3]**, Lin Liu**[3]**, Dacheng Tao**[2]**, Liangpei Zhang**[4][†]

[1]School of Computer Science, National Engineering Research Center for Multimedia Software, Institute of Artificial Intelligence, and Hubei Key Laboratory of Multimedia and Network Communication Engineering, Wuhan University, China
[2]School of Computer Science, Faculty of Engineering, The University of Sydney, Australia
[3]National Engineering Research Center of Speech and Language Information Processing, China
[4]State Key Laboratory of Information Engineering in Surveying, Mapping and Remote Sensing, Wuhan University, China
`{d_wang,dubo,zlp62}@whu.edu.cn; jing.zhang1@sydney.edu.au;`
`{mqxu7,linliu}@iflytek.com; dacheng.tao@gmail.com`

## Abstract

The success of the Segment Anything Model (SAM) demonstrates the significance of data-centric machine learning. However, due to the difficulties and high costs associated with annotating Remote Sensing (RS) images, a large amount of valuable RS data remains unlabeled, particularly at the pixel level. In this study, we leverage SAM and existing RS object detection datasets to develop an efficient pipeline for generating a large-scale RS segmentation dataset, dubbed SAMRS. SAMRS totally possesses 105,090 images and 1,668,241 instances, surpassing existing high-resolution RS segmentation datasets in size by several orders of magnitude. It provides object category, location, and instance information that can be used for semantic segmentation, instance segmentation, and object detection, either individually or in combination. We also provide a comprehensive analysis of SAMRS from various aspects. Moreover, preliminary experiments highlight the importance of conducting segmentation pre-training with SAMRS to address task discrepancies and alleviate the limitations posed by limited training data during fine-tuning. The code and dataset will be available at SAMRS.

## 1 Introduction

The advancement of earth observation technologies has led to the generation of abundant remote sensing images (RSI). These images retain valuable information about the spatial distribution and condition of extensive ground surfaces and geospatial objects, and can be conveniently accessed in real-time. Consequently, remote sensing data has garnered the interest of various disciplines, including agricultural monitoring, urban planning, and environmental protection. In particular, the identification of surface targets has been a fundamental task in these fields for several years.

To our knowledge, a significant number of RSIs remain unlabeled. Unlike natural images that can be easily comprehended by the human eye, interpreting RSI taken from an aerial perspective typically demands specialized expertise from practitioners. Furthermore, RSI objects are often distributed sparsely, and the images frequently contain small targets, making the labeling process less efficient. Therefore, the annotation of RSI has traditionally required substantial labor and time costs. Among

---

[*]This work was partially done during Di Wang's internship at iFlytek.
[†]Corresponding author.

37th Conference on Neural Information Processing Systems (NeurIPS 2023) Track on Datasets and Benchmarks.

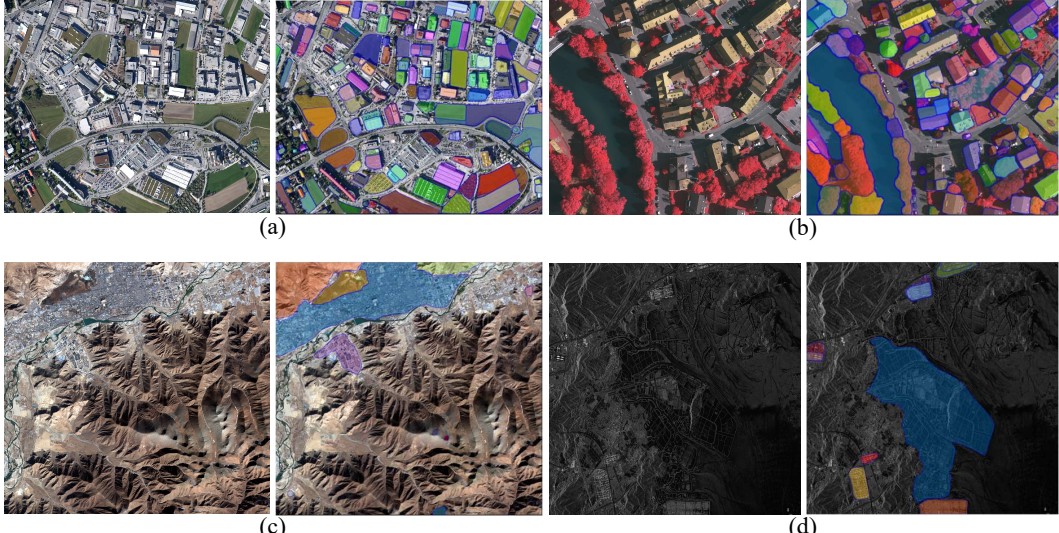

Figure 1: Some examples of SAM segmentation results on RSIs: (a) RGB aerial image obtained from the IsAID dataset [41]. (b) Airborne aerial image composed of near-infrared, red, and green bands. This image is from the ISPRS Vaihingen dataset[1]. (c) RGB satellite image observed by GF-2 sensors. This image is from the GID dataset [34]. (d) Hisea-1 SAR image from the Marine Farms Segmentation track of the 5th Gaofen Challenge[2]. These segmentation results are generated by the SAM demo website[3].

various RS tasks, the classification task requires only a single category for the entire scene, and the detection task involves the additional step of bounding box annotation, while segmentation is particularly challenging since it necessitates pixel-level annotations to accurately delineate object boundaries.

Do we have to spend a significant amount of time annotating RSIs? The answer is probably no. Recently, the segment anything model (SAM) [17], which excels in object segmentation, has gained popularity as a new research focus in the field of computer vision. SAM accurately captures object locations and contours (*i.e.*, in the form of masks), enabling it to distinguish various objects in the foreground and background. Furthermore, SAM possesses an impressive zero-shot segmentation ability, exhibiting high performance even when applied to specialized scenarios such as cell images photographed by microscopes [8] and medical images [26], despite being trained on a vast dataset of natural images. In the RS field, [31] firstly tests the performance of SAM on six public datasets. [16] extra introduce a domain decoder to improve the performance of SAM on the planetary geological mapping task. Beyond default prompts, [29, 45] consider utilizing texts as the prompt by adopting Grounding DINO [20] to obtain boxes that can be employed by SAM. Then, [29] realizes the one-shot segmentation with the help of PerSAM [47], while [45] applies the heatmap obtained from CLIP [30] to further optimize segmentation results. Different from the above methods with manual prompts, [3] design a prompter to adaptively generate prompts for improving the performance of SAM in instance segmentation. In addition, SAM is also used in producing rotated bounding boxes [4], which is significant for RS oriented object detection.

We have also found it performs well in recognizing diverse targets in RSI, even when the images are obtained using sensors that perceive different bands, such as infrared and microwave, or with varying resolutions, such as airborne or satellite imagery, as illustrated in Figure 1. Although we acknowledge that SAM may not have fully detected all regions, we believe that it has significant potential to improve the efficiency of annotating RSIs since it delivers promising segmentations on

---

[1]https://www.isprs.org/education/benchmarks/UrbanSemLab/2d-sem-label-vaihingen.aspx
[2]https://www.gaofen-challenge.com/challenge
[3]https://segment-anything.com/demo
[4]https://github.com/Li-Qingyun/sam-mmrotate

recognized areas. Therefore, in this study, we aim to utilize SAM to efficiently construct a large-scale RS segmentation dataset by obtaining pixel-level annotations for RSIs. Ground objects in RSI possess definite category properties, which are essential for real RS recognition tasks. However, the segmentation maps produced by SAM lack such information, rendering them unsuitable for labeling RSIs. To address this issue, we notice the annotations in existing RS object detection datasets, which include category and bounding box information. With the aid of SAM, we can leverage such detection annotations to obtain pixel-level semantic labels and efficiently construct large-scale segmentation datasets. The obtained dataset is called **S**egment **A**nything **M**odel annotated **R**emote Sensing **S**egmentation dataset (SAMRS). SAMRS inherits the characteristics of existing RS object detection datasets that have more samples and categories compared with existing high-resolution RS segmentation datasets.

Since we efficiently obtain numerous segmentation label maps, it is natural to consider using the obtained dataset for pre-training. Existing models pretrained by classification tasks may be not very suitable for downstream tasks, e.g., segmentation, because of the task-level discrepancy [36], while the emergence of SAMRS is expected to address this issue. To this end, we train classical deep learning models on the SAMRS, and finetune the trained model on typical RS segmentation datasets to explore the feasibility of segmentation pre-training. The main contribution of this study can be summarized to: **(1)** We develop a SAM-based pipeline for efficiently generating RS segmentation annotations. **(2)** We obtain a large-scale RS segmentation dataset named SAMRS using existing RS object detection annotations, whose capacity is far beyond existing high-resolution RS segmentation datasets. **(3)** We conduct preliminary segmentation pre-training experiments on SAMRS. The results highlight the importance of conducting segmentation pre-training using large-scale RS segmentation data, such as SAMRS, for mitigating task discrepancy and dealing with limited training data. We hope this research could significantly enhance the annotation efficiency of RSIs, thereby unlocking the full potential of RS models, especially in the context of segmentation tasks.

## 2   Implementation

### 2.1   Segment Anything Model

To perform segmentation, additional prompts are needed to guide SAM to locate the object of interest, in addition to the input image. SAM supports various prompts, such as points, boxes, and masks, which can be input into the model either alone or in combination. It is important to note that when using point prompts, it is necessary to indicate whether the points are foregrounds or backgrounds. In this study, we use detection annotations from existing datasets to obtain all kinds of prompts since they contain both location and category information.

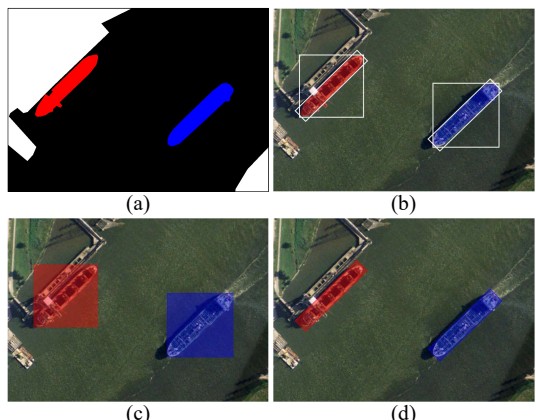

Figure 2: The differences between segmentation labels and mask prompts. (a) Pixel-level annotated map from the original dataset. (b) Pixel-level annotations along with horizontal and rotated box ground truths. (c) Mask prompts derived from horizontal boxes. (d) Mask prompts derived from rotated boxes. The ship instances are marked with different colors by following (a).

### 2.2   Datasets

In this study, we employ SAM on four public RS object detection datasets, namely HRSC2016 [22], DOTA-V2.0 [10], DIOR [18], and FAIR1M-2.0 [33]. DOTA, DIOR, and FAIR1M are three large-scale datasets [33]. HRSC2016 is primarily designed for ship detection and comprises only one category. In comparison to the other three datasets, it has the smallest data volume. Additionally, in the testing set, 124 images possess bounding box annotations and pixel-level labels simultaneously, making it highly suitable for evaluating the accuracy of SAM annotations. Therefore, we conduct an ablation study on the testing set consisting of the aforementioned 124 images to determine the optimal configuration for SAM. Following this, we generate segmentation labels for the remaining datasets. To obtain a segmentation dataset with more images or

categories, we opt for the latest versions of DOTA and FAIR1M. Based on the available annotations, we only transform the training and validation sets of DOTA-V2.0 and FAIR1M-2.0, while for DIOR, all data has been utilized. Here, according to the licenses, DOTA, DIOR, and FAIR1M can be used for academic purposes.

## 2.3 Prompt Settings

As RSIs are captured from an overhead perspective, the objects in them can have arbitrary orientations, unlike natural image objects that are typically oriented upward due to gravity. Hence, in addition to the usual horizontal bounding boxes (H-Box), we also consider oriented bounding boxes or rotated bounding boxes (R-Box) as box prompts. However, SAM does not directly support R-Box prompts. To address this issue, we use the minimum circumscribed horizontal rectangle of the R-Box, which is denoted as "RH-Box". It is also worth noting that the instances in the HRSC2016 testing set contain both H-Box and R-Box ground truth annotations.

In the case of the point prompt, due to the intricate shapes of various RS objects, such as airplanes, we have taken a cautious approach and only consider the center point as the foreground. We did not include background points in our study, as accurately defining them in an automated way can be challenging without additional contextual information. Regarding the mask prompt, we define the region enclosed by corresponding boxes as the mask prompt. Figure 2 illustrates the differences between the adopted mask prompts and ground truth segmentation labels. In SAM, the mask is a single-channel score matrix where positive values denote the active area where the target is located, whereas negative values represent irrelevant areas. In our experiments, we assign the values in these two types of areas as 1,000 and -1,000, respectively.

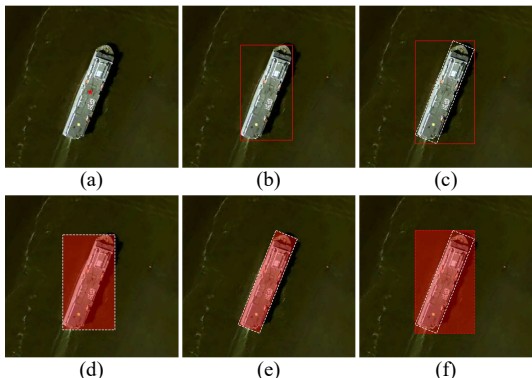

Figure 3: The adopted basic prompts. (a) CP. (b) H-Box. (c) RH-Box. (d) H-Box-M. (e) R-Box-M. (f) RH-Box-M. The dashed line is used for the convenience of visualization.

In summary, we have obtained six basic prompts, namely center point (CP), H-Box, RH-Box, and their corresponding masks, *i.e.*, H-Box-M, R-Box-M, and RH-Box-M, as illustrated in Figure 3.

## 2.4 Ablation Study

In addition to the above basic prompts, we also investigate various combinations of prompts in this study. To conduct a comprehensive analysis, we compute two types of mean intersection over union (mIOU) metrics: $\text{mIOU}_I$ and $\text{mIOU}_P$, which measure the similarity between the predicted segmentation mask and the ground truth label. The former is the average value of the IoU calculated on a per-instance basis, while the latter measures the pixel-level accuracy. Given the $i$th instance with intersection set $I_i$ and union set $U_i$, and the number of instances $N$, we have:

$$\text{mIOU}_I = \frac{1}{N}\sum_{i=1}^{N}\frac{I_i}{U_i} \quad \text{mIOU}_P = \frac{\sum_{i=1}^{N}I_i}{\sum_{i=1}^{N}U_i}. \tag{1}$$

Table 1 presents the evaluation results of utilizing different prompts. The point prompt delivers the worst performance and negatively affects the accuracy of any prompt combinations. This could be attributed to the insufficient amount of foreground points, which cannot guide the model effectively. The mask prompt performs better than the point prompt, but it still cannot generate high-quality segmentation annotations. The highest accuracy achieved by a mask prompt is approximately 60%, which is still much lower than the optimal prompts. Furthermore, the mask prompt has a negative impact on the performance of box prompts. When solely adopting the H-Box prompt, we obtain the highest accuracy compared to the point and mask prompts. For the case of utilizing R-Box annotations, the RH-Box prompt also achieves satisfactory performance. From this experiment, we conclude that: *if an RS object detection dataset only has R-Box annotations, then the RH-Box prompt*

Table 1: Results of using different prompts on the HRSC2016 testing set consisting of 124 images.

| CP | H-Box | H-Box-M | R-Box-M | RH-Box | RH-Box-M | mIOU$_I$ | mIOU$_P$ |
|---|---|---|---|---|---|---|---|
| *Point* | | | | | | | |
| ✓ | | | | | | 16.14 | 2.72 |
| *H-Box* | | | | | | | |
| | ✓ | | | | | **89.97** | **79.40** |
| | | ✓ | | | | 40.54 | 36.71 |
| ✓ | ✓ | | | | | 86.67 | 77.35 |
| | ✓ | ✓ | | | | 74.21 | 62.25 |
| ✓ | | ✓ | | | | 24.54 | 5.41 |
| ✓ | ✓ | ✓ | | | | 59.71 | 49.30 |
| *R-Box* | | | | | | | |
| | | | ✓ | | | **65.54** | **59.78** |
| ✓ | | | ✓ | | | 26.49 | 4.97 |
| *RH-Box* | | | | | | | |
| | | | | ✓ | | **88.85** | **76.42** |
| | | | | | ✓ | 34.63 | 31.81 |
| ✓ | | | | ✓ | | 83.55 | 72.67 |
| | | | | ✓ | ✓ | 66.23 | 52.75 |
| ✓ | | | | | ✓ | 23.71 | 5.10 |
| ✓ | | | | ✓ | ✓ | 49.24 | 39.03 |

Table 2: Comparisons of different high-resolution RS segmentation datasets.

| Dataset | #Images | #Category | #Channels | Resolution (m) | Image size | Instance | Fine-grained |
|---|---|---|---|---|---|---|---|
| ISPRS Vaihingen [1] | 33 | 6 | IR,R,G | 0.09 | 2,494 × 2,064 | | |
| ISPRS Potsdam [5] | 38 | 6 | IR,RGB | 0.05 | 6,000 × 6,000 | | |
| Zurich Summer [35] | 20 | 8 | NIR,RGB | 0.62 | 1,000 × 1,150 | | |
| Zeebruges [27] | 7 | 8 | RGB | 0.05 | 10,000 × 10,000 | | |
| DeepGlobe Land Cover [6] | 1,146 | 7 | RGB | 0.5 | 2,448 × 2,448 | | |
| UAVid [24] | 420 | 8 | RGB | - | 4,096 × 2,160 or 3,840 × 2,160 | | |
| GID [34] | 150 | 15 | NIR,RGB | 1 or 4 | 6,800 × 7,200 | | |
| Landcover.ai [2] | 41 | 3 | RGB | 0.25 or 0.5 | 9,000 × 9,500 or 4,200 × 4,700 | | |
| IsAID [41] | 2,806 | 15 | RGB | - | 800 × 800 ∼ 4,000 × 13,000 | ✓ | |
| LoveDA [38] | 5,987 | 7 | RGB | 0.3 | 1,024 × 1,024 | | |
| **SAMRS** | | | | | | | |
| **SOTA** | 17,480 | 18 | RGB | - | 1,024 × 1,024 | ✓ | |
| **SIOR** | 23,463 [1] | 20 | RGB | - | 800 × 800 | ✓ | |
| **FAST** | 64,147 | 37 | RGB | - | 600 × 600 | ✓ | ✓ |

[1] To avoid data snooping, only the 11725 images corresponding to the original DIOR trainval dataset are used in subsequent pre-trainings.

*should be used; otherwise, the H-Box prompt should be adopted.* This consideration is applied in our later dataset transformations.

## 2.5 Dataset Transformation

For the FAIR1M-2.0 dataset, since it only contains R-Box annotations, we use the corresponding RH-Box as the prompt. For DOTA-V2.0 and DIOR, we directly adopt the H-Box prompt. Prior to transformation, we follow the common practice to crop images in DOTA and FAIR1M datasets to 1,024 × 1,024 and 600 × 600, respectively, while images in DIOR are maintained at the size of 800 × 800. The resulting datasets are named SOTA (*i.e.*, DOTA → SOTA), SIOR (*i.e.*, DIOR → SIOR), and FAST (*i.e.*, Fine-grAined object recognItion in high-Resolution remote sensing imagery → Fine-grAined Segmentation for high-resolution remoTe sensing imagery), respectively. These datasets constitute a comprehensive and large-scale remote sensing segmentation database, *i.e.*, **SAMRS**.

## 3 SAMRS

### 3.1 Basic Information

The obtained segmentation labels are stored in *.png files. Pixel values are aligned with the object classes of source object detection datasets. The areas that have not been covered by the generated masks will be in a pixel value of 255. We present the comparison of our SAMRS dataset with existing high-resolution RS segmentation datasets in Table 2 from different aspects. With the available high-resolution RSI object detection datasets, we can efficiently annotate 105,090 images containing 1,668,241 instances based on SAM and the identified prompt settings (Sec. 2.4), which is more than ten times the capacity of existing datasets. Additionally, SAMRS inherits the categories

[5] https://www.isprs.org/education/benchmarks/UrbanSemLab/2d-sem-label-potsdam.aspx

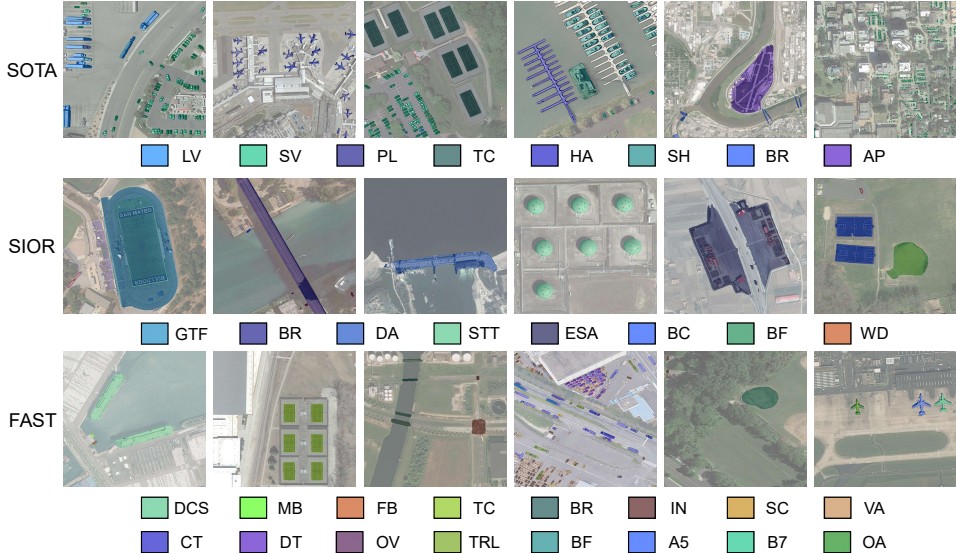

Figure 4: Some visual examples from the three subsets of our SAMRS dataset. For the definition of classes, please refer to the supplementary material.

of the original detection datasets, which makes them more diverse than other high-resolution RS segmentation collections. It is worth noting that RS object datasets usually have more diverse categories than RS segmentation datasets due to the difficulty of tagging pixels in RSIs, and thus our SAMRS reduces this gap.

Specifically, the resulting FAST dataset is a large-scale fine-grained RS segmentation dataset that targets diverse vehicles and grounds, while SOTA and SIOR are segmentation datasets containing common object categories. For this reason, we did not unify their categories. In addition to the massive pixel-level semantic mask annotations, SAMRS includes instance mask and bounding box annotations. This means that *it can be used to perform semantic segmentation, instance segmentation, and object detection, either individually or in combination.* This feature sets SAMRS apart from the IsAID dataset, which was independently annotated from scratch on DOTA-V1.0 [42] images.

## 3.2 Statistics and Analysis

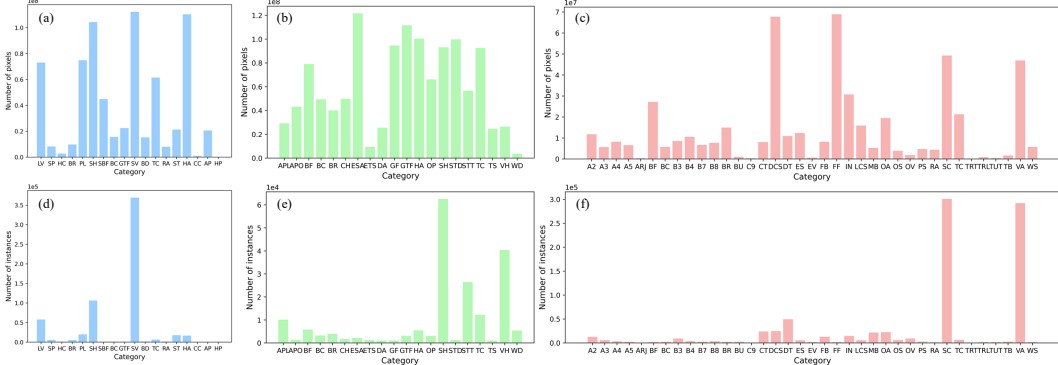

Figure 5: Statistics of the number of pixels and instances per category in SAMRS. The histograms for the subsets SOTA, SIOR, and FAST are shown in the first, second, and third columns, respectively. The first row presents histograms on a per-pixel basis, while the second row presents histograms on a per-instance basis. A list of category abbreviations is provided in the supplementary material.

To gain a deeper understanding of the characteristics of the SAMRS dataset, we conduct a thorough analysis of their capacity per category, including pixel and instance numbers. The results are presented

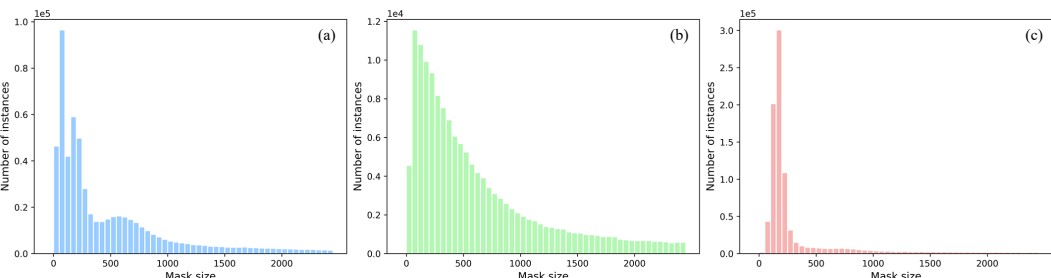

Figure 6: Statistics of the mask sizes in the subsets of SAMRS. (a) SOTA. (b) SIOR. (c) FAST.

in Figure 5. In this analysis, we only count instances that have valid masks. The figure indicates that SIOR has more balanced categories compared to SOTA and FAST. In the instance-level statistics, we observe a large number of vehicle annotations, particularly on small ships and cars, as they are common in the real world and frequently appear in RSIs. This could also be the goal of initially developing these detection datasets. For instance, DOTA-V2.0 focuses on small targets, while FAIR1M mainly aims to accurately distinguish between different types of vehicles. Furthermore, it is observed that some categories have a high number of pixels but a low number of instances, which is likely due to their large size. For instance, the *expressway-service-area* in SIOR and the *football-field* in FAST demonstrate this pattern.

In addition, we investigate the distribution of mask sizes in SAMRS, as shown in Figure 6. The results indicate that, in general, there are more instances with smaller sizes in all subsets. However, some differences exist between the subsets. Specifically, FAST has more small objects than the other two sets. Nevertheless, SOTA appears to have a higher number of extremely small targets (*i.e.*, <100 pixels), since its source dataset DOTA-V2.0 is designed for small object detection. On the other hand, SIOR has a more smooth distribution of mask sizes compared to SOTA and FAST.

### 3.3 Visualization

In Figure 4, we visualize some segmentation annotations from the three subsets in our SAMRS dataset. As can be seen, SOTA exhibits a greater number of instances for tiny cars, whereas FAST provides a more fine-grained annotation of existing categories in SOTA such as car, ship, and plane. SIOR on the other hand, offers annotations for more diverse ground objects, such as *dam*. Hence, our SAMRS dataset encompasses a wide range of categories with varying sizes and distributions, thereby presenting a new challenge for RS semantic segmentation.

## 4 Experiment

### 4.1 Pre-training

#### 4.1.1 Data and Model Settings

To investigate the influence of segmentation pre-training (SEP) using SAMRS, we adopt multiple segmentation frameworks, including typical encoder-decoder networks and the recently emerged end-to-end structure. In encoder-decoder networks, we utilize classical UNet [32] and commonly-used UperNet [43]. Different from the original U-Net that has five blocks in the decoder part, to be compatible with the typical hierarchical pyramid backbone network that outputs four levels of features, we replace the last block to a single $2\times$ bilinear upsampling layer, which is followed by a segmentation head that contains a $1\times1$ convolution, a $2\times$ bilinear upsampling, and a ReLU activation function. For UperNet, the segmentation head only employs a $1\times1$ convolution. To comprehensively explore the SEP, in addition to traditional convolutional networks such as ResNet [14], diverse backbones are used, including hierarchical vision transformers: Swin [21], ViTAEv2 [46] and InternImage [40], and non-hierarchical networks, including ViT [12], ViT-Adapter [4] and ViT-RVSA [37]. For the end-to-end structure, we choose the recent Mask2Former [5]. The SAMRS is split into two parts, one for pre-training, and another for validation, see the supplementary material for more details. In the data preprocessing, we employed common data augmentation techniques,

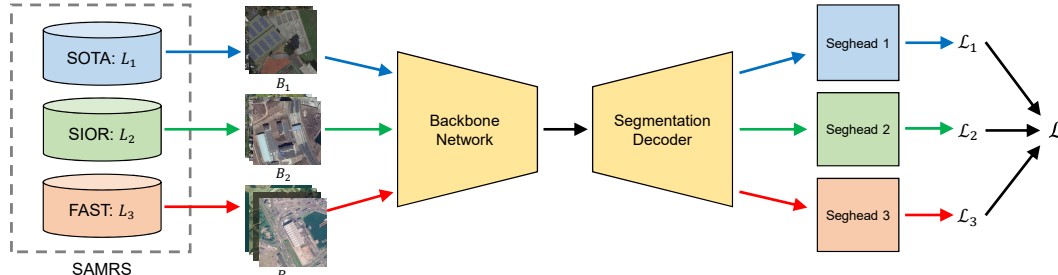

Figure 7: The pipeline of segmentation pre-training on SAMRS. Different colors represent the data stream of various sets. The yellow parts will be used in fine-tuning.

including random scaling, random horizontal and vertical flipping, random rotation by 90 degrees, and altering pixel values through random color jitter and gamma transformation. Moreover, to ensure a fair comparison with prior studies [7, 36], we randomly cropped the input images to a size of 224 × 224.

### 4.1.2 Training Settings

Intuitively, a well-initialized backbone network generates discriminative features at the beginning of training, thereby facilitating the optimization process of the decoder component. Since the SEP is expected to mitigate the gaps between pre-training and downstream tasks. To this end, before the segmentation pre-training phase, the selected model's backbone network is initialized using pretrained weights. In our experiments, to fully evaluate the SEP, besides basic supervised pre-training on ImageNet (IMP) [7], we also utilize the RSP [36] on the MillionAID Dataset [23]. In addition, the unsupervised pre-training weights are also involved, including BEiT [1] and MAE [13]. Here, the MAE pre-training is conducted on the MillionAID [37], while the BEiT is pretrained on the ImageNet.

To accommodate the multiple segmentation sets within SAMRS, each having a different number of categories, we employ a multi-head pre-training strategy. This approach involves utilizing separate segmentation heads for individual datasets. The only distinction lies in the output channel count of the $1 \times 1$ convolution, which corresponds to the number of categories. During batch-based training, diverse mini-batches are sampled from these sets to form a collective large batch, which is then fed into the network. Given the volume disparities among the various sets in SAMRS, proportional sampling is employed to obtain the mini-batches. Assuming a large batch of size $B$ consists of $M$ mini-batches with sizes $B_1, B_2, \cdots, B_M$, it follows that $B = B_1 + B_2 + \cdots + B_M$. Each mini-batch corresponds to its respective segmentation head, resulting in a training loss $\mathcal{L}_i$ for the $i$th mini-batch. The total loss is computed as $\mathcal{L} = \mathcal{L}_1 + \mathcal{L}_2 + \cdots + \mathcal{L}_M$. Assuming the sizes of the $M$ sets in SAMRS are $L_1, L_2, \cdots, L_M$, we can express this relationship as $B_i = \frac{L_i}{\sum_{j=1}^{M} L_j} B$, $i = 0, 1, ..., M$. Here, $M$ can be easily extended for more RS detection datasets. In this study, $M = 3$. Figure 7 illustrates the pre-training pipeline. Each model is first pre-trained for 80k iterations with $B = 96$ and then used for fine-tuning. All experiments are implemented by PyTorch on NVIDIA GeForce RTX 3090 GPUs.

### 4.2 Fine-tuning

#### 4.2.1 Comparison to Various Pre-training Strategies

In the RS community, ISPRS Potsdam and IsAID are commonly-used finely annotated datasets for evaluating segmentation methods [25, 28, 36, 37, 44], and we use them to assess the pre-trained models, as Table 3-4 shown. It can be seen that, without good initialization, the performances of SEP are comparable as IMP but inferior to IMP+SEP. On the Potsdam scene, for the traditional encoder-decoder network, SEP improves both convolutional and vision transformer networks, especially for UperNet and the backbones containing hierarchical features (also includes ResNet). As a result, ViTAEv2-S is greatly boosted and overperforms existing advanced methods in overall accuracy. We also observe that SEP is useful when combined with different pre-training strategies. Even if using the initialized weights generated by pre-training on SAMRS itself, SEP still can improve the accuracy,

Table 3: Segmentation results of different methods on the ISPRS Potsdam dataset. ‡: MAE pre-training on the MillionAID. ‡‡: MAE pre-training on the SAMRS training set. "*" denotes the best score among all methods.

| Method | Pretrain | Backbone | F1 score per category | | | | | OA | mF1 |
|---|---|---|---|---|---|---|---|---|---|
| | | | Imper. surf. | Building | Low veg. | Tree | Car | | |
| *Comparison method* | | | | | | | | | |
| ST-UNet [15] | — | ResNet-50 | 79.19 | 86.63 | 67.89 | 66.37 | 79.77 | — | 86.13 |
| ResUNet-a d7v2 [9] | — | — | 93.50 | 97.20 | 88.20 | 89.20 | 96.40 | 91.50 | 92.90 |
| LANet [11] | IMP | ResNet-50 | 93.05 | 97.19 | 87.30 | 88.04 | 94.19 | 90.84 | 91.95 |
| DCFAM [39] | IMP | Swin-S | 94.19 | 97.57 | 88.57 | 89.62 | 96.31 | 92.00 | 93.25* |
| *Convolutional network* | | | | | | | | | |
| UNet | SEP | ResNet-50 | 90.62 | 94.75 | 85.12 | 83.91 | 96.51 | 89.70 | 90.18 |
| UNet | IMP | ResNet-50 | 90.78 | 94.78 | 85.23 | 84.76 | 96.81 | 89.94 | 90.47 |
| UNet | IMP+SEP | ResNet-50 | 91.36 | 94.92 | 85.39 | 85.24 | 97.17 | **90.29** | **90.82** |
| UperNet | SEP | ResNet-50 | 91.02 | 94.82 | 84.28 | 83.97 | 96.95 | 89.70 | 90.21 |
| UperNet | IMP | ResNet-50 | 90.70 | 94.44 | 84.68 | 83.94 | 96.58 | 89.59 | 90.07 |
| UperNet | IMP+SEP | ResNet-50 | 91.38 | 95.26 | 85.14 | 84.88 | 97.16 | **90.27** | **90.76** |
| *Hierarchical vision transformer* | | | | | | | | | |
| UperNet | IMP | Swin-T | 93.09 | 96.74 | 86.99 | 86.45 | 91.12 | 91.44 | 90.88 |
| UperNet | IMP+SEP | Swin-T | 93.06 | 96.65 | 87.07 | 86.74 | 97.64 | **91.88** | **92.23** |
| UperNet | IMP | ViTAEv2-S | 92.54 | 96.54 | 86.11 | 86.13 | 91.31 | 91.00 | 90.52 |
| UperNet | IMP+SEP | ViTAEv2-S | 93.45 | 96.99 | 87.65 | 87.00 | 97.67 | **92.25*** | **92.55** |
| UperNet | IMP | InternImage-T | 93.27 | 96.80 | 87.41 | 86.62 | 91.79 | 91.65 | 91.18 |
| UperNet | IMP+SEP | InternImage-T | 93.30 | 96.91 | 87.24 | 86.80 | 97.81 | **92.08** | **92.41** |
| *Plain vision transformer* | | | | | | | | | |
| UperNet | IMP | ViT-B | 93.09 | 96.83 | 86.93 | 86.61 | 90.93 | 91.47 | 90.88 |
| UperNet | IMP+SEP | ViT-B | 92.96 | 96.52 | 86.62 | 86.01 | 97.57 | **91.60** | **91.94** |
| UperNet | IMP | ViT-Adapter-B | 93.16 | 96.77 | 87.09 | 86.71 | 91.20 | 91.53 | 90.98 |
| UperNet | IMP+SEP | ViT-Adapter-B | 93.20 | 96.75 | 87.06 | 86.52 | 97.68 | **91.91** | **92.24** |
| *Pre-training strategy* | | | | | | | | | |
| UNet | RSP | ResNet-50 | 91.49 | 95.42 | 85.70 | 85.18 | 97.05 | 90.49 | 90.97 |
| UNet | RSP+SEP | ResNet-50 | 92.00 | 95.44 | 85.76 | 85.33 | 97.38 | **90.72** | **91.18** |
| UperNet | RSP | ResNet-50 | 91.08 | 94.64 | 85.57 | 85.38 | 96.97 | 90.18 | 90.73 |
| UperNet | RSP+SEP | ResNet-50 | 91.73 | 95.52 | 85.44 | 85.35 | 97.24 | **90.59** | **91.06** |
| UperNet | BEiT | ViT-B | 88.70 | 92.29 | 81.48 | 78.64 | 96.36 | 86.86 | 87.49 |
| UperNet | BEiT+SEP | ViT-B | 89.95 | 93.33 | 82.96 | 80.91 | 96.67 | **88.20** | **88.76** |
| UperNet | MAE ‡ | ViT-B + RVSA | 92.67 | 96.38 | 86.43 | 85.89 | 90.46 | 90.97 | 90.37 |
| UperNet | MAE+SEP | ViT-B + RVSA | 92.69 | 96.33 | 86.28 | 85.60 | 97.56 | **91.33** | **91.69** |
| UperNet | SAMRS-MAE ‡‡ | ViT-B + RVSA | 92.46 | 96.10 | 86.18 | 85.59 | 90.35 | 90.71 | 90.13 |
| UperNet | SAMRS-MAE+SEP | ViT-B + RVSA | 92.34 | 95.88 | 86.06 | 85.32 | 97.54 | **91.01** | **91.43** |
| *End-to-end transformer* | | | | | | | | | |
| Mask2Former | IMP | ResNet-50 | 88.40 | 92.93 | 83.05 | 83.98 | 86.00 | **87.54** | **86.87** |
| Mask2Former | IMP+SEP | ResNet-50 | 72.41 | 78.98 | 63.14 | 61.62 | 73.16 | 70.14 | 69.86 |

excluding the effect of data volume used for training. We notice SEP played a negative role in the end-to-end structure, it may be because the objects of SAMRS are too small, which is unfavorable for the region-based Mask2Former. In addition, the Mask2Former, which has obtained high accuracies on natural images, does not perform as well as UNet and UperNet on RSIs. These results indicate more refined parameter adjustments of Mask2Former are needed in later research. On the IsAID dataset, the performances of SEP on simple convolutional networks depending on local perception are unstable, because IsAID and DOTA share the same images but with different annotations, which may confuse the model. Benefiting from SEP, vision transformer networks are further enhanced and surpass previous methods. From these results we can see, SEP is able to mitigate the influence of task-level disparities, specifically the gaps between upstream pre-training tasks and downstream segmentation tasks.

## 4.3 Fine-tuning with Small-size Training Samples

The difficulty of annotating pixel-level masks limits the scale of existing remote sensing segmentation datasets, ultimately constraining the performance of trained models due to insufficient training samples. In order to investigate the effectiveness of SEP under conditions of limited training samples, we conducted experiments involving fine-tuning models using small fractions (1%, 3%, and 5%) of data from the ISPRS Potsdam and IsAID training sets, as outlined in Table 5 and Table 6. The integration of SEP with SAMRS, which provides a valuable segmentation prior, yields superior results compared to the IMP and RSP counterparts. This advantage is particularly evident when the number of available samples is scarce in the ISPRS Potsdam scene. Conversely, the results for the IsAID dataset exhibit an opposite trend due to the inherent challenges of this dataset, where both IMP and RSP yield extremely low overall accuracies. Nevertheless, the adoption of SEP significantly improves model performance. These findings highlight the importance of conducting segmentation

Table 4: Segmentation results of different methods on the IsAID dataset. ‡: MAE pre-training on the MillionAID. "*" denotes the best score among all methods.

| Method | Pretrain | Backbone | IOU per category[1] | | | | | | | | | | | | | | | mIOU |
|---|---|---|---|---|---|---|---|---|---|---|---|---|---|---|---|---|---|---|
| | | | SH | ST | BD | TC | BC | GTF | BR | LV | SV | HC | SP | RA | SBF | PL | HA | |
| *Comparison method* | | | | | | | | | | | | | | | | | | |
| HMANet [28] | IMP | ResNet-50 | 65.38 | 70.92 | 74.71 | 88.69 | 60.51 | 54.57 | 28.98 | 59.74 | 50.28 | 32.58 | 51.41 | 62.88 | 70.20 | 83.79 | 51.91 | 62.64 |
| UperNet | MAE ‡ | ViTAE-B + RVSA◇ | 70.61 | 77.19 | 73.14 | 70.53 | 53.46 | 59.69 | 44.78 | 63.89 | 52.56 | 38.05 | 47.07 | 66.51 | 74.61 | 85.78 | 54.90 | 64.49 |
| FactSeg [25] | IMP | ResNet-50 | 68.34 | 56.83 | 78.36 | 88.91 | 64.89 | 54.60 | 36.34 | 62.65 | 49.53 | 42.72 | 51.47 | 69.42 | 73.55 | 84.13 | 55.74 | 64.79 |
| RssFormer [44] | IMP | RSS-B | — | — | — | — | — | — | — | — | — | — | — | — | — | — | — | 65.88 |
| *Convolutional network* | | | | | | | | | | | | | | | | | | |
| UNet | IMP | ResNet-50 | 50.11 | 48.50 | 31.52 | 74.30 | 15.38 | 14.99 | 8.83 | 51.59 | 34.71 | 0.07 | 39.03 | 7.88 | 39.33 | 73.12 | 47.71 | 35.80 |
| UNet | IMP+SEP | ResNet-50 | 62.01 | 62.58 | 66.26 | 85.84 | 51.65 | 37.77 | 31.99 | 61.03 | 45.57 | 3.14 | 45.61 | 56.29 | 64.40 | 81.58 | 54.22 | **54.00** |
| UNet | RSP | ResNet-50 | 53.00 | 53.53 | 62.99 | 78.25 | 34.48 | 36.92 | 24.83 | 54.95 | 34.22 | 2.11 | 40.36 | 42.18 | 50.37 | 77.82 | 48.81 | **46.32** |
| UNet | RSP+SEP | ResNet-50 | 46.83 | 28.82 | 26.58 | 67.30 | 13.83 | 17.79 | 3.97 | 48.37 | 28.30 | 0.00 | 35.30 | 0.00 | 28.74 | 67.62 | 43.29 | 30.45 |
| UperNet | IMP | ResNet-50 | 69.81 | 71.08 | 75.99 | 87.73 | 58.11 | 61.96 | 40.06 | 63.84 | 48.54 | 27.02 | 48.59 | 70.19 | 76.15 | 82.94 | 56.05 | **62.54** |
| UperNet | IMP+SEP | ResNet-50 | 66.37 | 65.80 | 70.24 | 86.60 | 54.35 | 47.23 | 31.71 | 63.46 | 46.25 | 22.26 | 48.01 | 61.29 | 72.28 | 81.22 | 54.44 | 58.10 |
| UperNet | RSP | ResNet-50 | 46.76 | 13.21 | 35.21 | 56.86 | 13.89 | 22.78 | 3.91 | 47.44 | 24.10 | 0.32 | 40.26 | 27.55 | 48.42 | 68.92 | 44.88 | 32.97 |
| UperNet | RSP+SEP | ResNet-50 | 66.62 | 65.37 | 72.49 | 86.88 | 57.18 | 50.20 | 37.71 | 63.29 | 47.41 | 22.96 | 45.36 | 65.98 | 74.91 | 77.97 | 54.53 | **59.26** |
| *Vision transformer* | | | | | | | | | | | | | | | | | | |
| UperNet | IMP | Swin-T | 71.26 | 73.97 | 77.54 | 87.33 | 56.42 | 61.53 | 42.36 | 65.89 | 49.74 | 38.30 | 47.82 | 66.08 | 76.22 | 84.72 | 57.39 | 63.77 |
| UperNet | IMP+SEP | Swin-T | 73.37 | 74.19 | 76.63 | 87.63 | 56.40 | 61.44 | 42.42 | 65.98 | 50.92 | 38.82 | 49.16 | 69.70 | 75.05 | 84.83 | 57.27 | **64.25** |
| UperNet | IMP | ViTAEv2-S | 73.31 | 76.22 | 75.12 | 89.65 | 62.72 | 65.71 | 42.94 | 67.09 | 52.01 | 39.56 | 48.07 | 71.38 | 79.04 | 85.64 | 58.47 | 65.79 |
| UperNet | IMP+SEP | ViTAEv2-S | 75.95 | 75.48 | 75.38 | 89.56 | 63.79 | 64.85 | 45.62 | 68.49 | 53.27 | 37.56 | 49.49 | 69.84 | 79.22 | 85.85 | 59.58 | **66.26\*** |
| UperNet | IMP | InternImage-T | 74.56 | 73.71 | 79.04 | 88.04 | 65.11 | 64.90 | 38.35 | 65.53 | 51.16 | 39.00 | 48.79 | 66.35 | 76.13 | 85.39 | 59.25 | 65.02 |
| UperNet | IMP+SEP | InternImage-T | 75.12 | 73.67 | 79.34 | 88.58 | 62.39 | 65.37 | 42.54 | 67.35 | 52.91 | 42.09 | 48.21 | 69.94 | 79.33 | 86.09 | 57.88 | **66.05** |

[1]  SH: ship. ST: storage tank. BD: baseball diamond. TC: tennis court. BC: baseball court. GTF: ground track field. BR: bridge. LV: large vehicle. SV: small vehicle. HC: helicopter. SP: swimming pool. RA: roundabout. SBF: soccer ball field. PL: plane. HA: harbor.

Table 5: Segmentation results of different pre-training methods on the ISPRS Potsdam dataset.

| Method | Pretrain | Backbone | mF1 | | |
|---|---|---|---|---|---|
| | | | 5% | 3% | 1% |
| UNet | IMP | ResNet-50 | 80.45 | 75.03 | 65.72 |
| UNet | IMP+SEP | ResNet-50 | 81.80 | 77.97 | 69.70 |
| | | Δ | +1.35 | +2.94 | +3.98 |
| UNet | RSP | ResNet-50 | 80.34 | 75.35 | 62.86 |
| UNet | RSP+SEP | ResNet-50 | 81.69 | 77.40 | 68.35 |
| | | Δ | +1.35 | +2.05 | +5.49 |

Table 6: Segmentation results of different pre-training methods on the IsAID dataset.

| Method | Pretrain | Backbone | mIOU | | |
|---|---|---|---|---|---|
| | | | 5% | 3% | 1% |
| UNet | IMP | ResNet-50 | 5.33 | 5.19 | 1.33 |
| UNet | IMP+SEP | ResNet-50 | 18.74 | 12.98 | 7.04 |
| | | Δ | +13.41 | +7.79 | +5.71 |
| UNet | RSP | ResNet-50 | 4.68 | 3.67 | 2.50 |
| UNet | RSP+SEP | ResNet-50 | 22.47 | 13.88 | 8.84 |
| | | Δ | +17.79 | +10.21 | +6.34 |

pre-training using large-scale RS segmentation data, such as SAMRS, prior to training with limited data. Notably, the developed pipeline enables the rapid construction of such a dataset at a low labeling cost, making it a promising approach.

### 4.4 Limitations and Discussion

Previous classical datasets, such as HRSC2016 [22] and COCO [19], simultaneously contain bounding box and pixel-level mask annotations, proving the feasibility of the coexistence for segmentation and detection labels. Therefore, it is reasonable to construct the SAMRS dataset by transforming existing RS object detection datasets. However, despite successfully establishing the SAMRS dataset, which outperforms existing high-resolution RS segmentation datasets by more than tenfold, its volume remains smaller than large-scale classification datasets such as ImageNet [7] and MillionAID [23], commonly employed for pre-training purposes. Our current investigation focuses exclusively on pre-training small-scale basic models (about 100M) and we intend to incorporate larger models. Additionally, it is also worth trying to explore the impact of pre-training on SAMRS for tasks such as instance segmentation and object detection.

## 5 Conclusion

This study presents an effective way to create a large-scale remote sensing (RS) segmentation dataset by harnessing the capabilities of the Segment Anything Model (SAM) and existing object detection datasets. Given the unique challenges associated with RS data labeling, we investigate the performance of various prompts to identify the optimal settings for SAM. By leveraging these optimal settings, we generate extensive mask annotations for RS images, thereby creating a large-scale segmentation dataset named SAMRS. Remarkably, SAMRS surpasses all previously available high-resolution RS segmentation datasets in terms of volume. Furthermore, our statistical analysis reveals that SAMRS encompasses a diverse array of categories exhibiting varying sizes and distributions. SAMRS can be utilized for semantic segmentation, instance segmentation, and object detection, either independently or in combination. Specifically, we present a preliminary investigation and demonstrate the value of segmentation pre-training on SAMRS for RS segmentation tasks, especially in scenarios with limited training samples.

## Acknowledgments

We acknowledge the authors of SAM for releasing codes and models, and the authors of DOTA, DIOR, and FAIR1M for providing their datasets. This work was supported in part by the National Natural Science Foundation of China under Grant 62225113 and in part by the National Key Research and Development Program of China under Grant 2022YFB3903405.

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
