# SAMRS: Scaling-up Remote Sensing Segmentation Dataset with Segment Anything Model: Supplementary Material

**Di Wang**[1]*, **Jing Zhang**[2]†, **Bo Du**[1]‡, **Minqiang Xu**[3], **Lin Liu**[3], **Dacheng Tao**[2], **Liangpei Zhang**[4]†

[1]School of Computer Science, National Engineering Research Center for Multimedia Software, Institute of Artificial Intelligence, and Hubei Key Laboratory of Multimedia and Network Communication Engineering, Wuhan University, China
[2]School of Computer Science, Faculty of Engineering, The University of Sydney, Australia
[3]National Engineering Research Center of Speech and Language Information Processing, China
[4]State Key Laboratory of Information Engineering in Surveying, Mapping and Remote Sensing, Wuhan University, China
{d_wang,dubo,zlp62}@whu.edu.cn; jing.zhang1@sydney.edu.au;
{mqxu7,linliu}@iflytek.com; dacheng.tao@gmail.com

## A  SAMRS

### A.1  Category Abbreviations

For the SOTA dataset, we present the list of all category abbreviations as follows. *LV: large vehicle, SP: swimming pool, HC: helicopter, BR: bridge, PL: plane, SH: ship, SBF: soccer ball field, BC: basketball court, GTF: ground track field, SV: small vehicle, BD: baseball diamond, TC: tennis court, RA: roundabout, ST: storage tank, HA: harbor, CC: container crane, AP: airport, HP: helipad.*

For the SIOR dataset, we present the list of all category abbreviations as follows. *APL: airplane, APO: airport, BF: baseball field, BC: basketball court, BR: bridge, CH: chimney, ESA: expressway service area, ETS: expressway toll station, DA: dam, GF: golf field, GTF: ground track field, HA: harbor, OP: overpass, SH: ship, STD: stadium, STT: storage tank, TC: tennis court, TS: train station, VH: vehicle, WD: windmill.*

For the FAST dataset, we present the list of all category abbreviations as follows. *A2: A220, A3: A321, A4: A330, A5: A350, ARJ: ARJ21, BF: baseball field, BC: basketball court, B3: boeing737, B4: boeing747, B7: boeing777, B8: boeing787, BR: bridge, BU: bus, C9: C919, CT: cargo truck, DCS: dry cargo ship, DT: dump truck, ES: engineering ship, EV: excavator, FB: fishing boat, FF: football field', IN: intersection, LCS: liquid cargo ship, MB: motorboat, OA: other airplane, OS: other ship, OV: other vehicle, PS: passenger ship, RA: roundabout, SC: small car, TC: tennis court, TRT: tractor, TRL: trailer, TUT: truck tractor, TB: tugboat, VA: van, WS: warship.*

## B  Pre-training and Fine-tuning

### B.1  Experiment Settings

We present the experiment settings of pre-training and fine-tuning in Table S1-S2.

---

*This work was partially done during Di Wang's internship at iFlytek.
†Corresponding author.

37th Conference on Neural Information Processing Systems (NeurIPS 2023) Track on Datasets and Benchmarks.

Table S1: Basic settings in experiments. "|" means pre-training | fine-tuning.

| Config | value |
|---|---|
| Optimizer | AdamW |
| Momentum | (0.9, 0.999) |
| Batchsize | 96 | 8 |
| Iterations | 80000 |
| Scheduler | cosine decay |

Table S2: Detailed settings of different models. "|" means pre-training | fine-tuning. ILR: Initial learning rate. WD: Weight decay. MLR: Minimum learning rate

| Backbone | ILR | WD | MLR |
|---|---|---|---|
| ResNet [9] | 1e-3 | 5e-2 | 1e-4 | 5e-6 |
| Swin [11] /ViTAEv2 [19] | 6e-5 | 1e-2 | 0 |
| InternImage [16] | 6e-5 | 5e-2 | 0 |
| ViT [7] /ViT-RVSA [15] | 6e-5 | 5e-2 | 0 |
| ViT-Adapter [2] | 6e-5 | 1e-2 | 0 |

## B.2 SAMRS Training and Validation sets

For the experiments based on the SAMRS dataset (see Table 2 in the main text), in each subset, 95% samples are used for pre-training. They together consist of the SAMRS training set. The remained samples consist of the SAMRS validation set. SAMRS training and validation sets have 88,685 and 4,667 images, respectively. The samples transformed from the DIOR testing set [10] have not been used in any experiment.

## B.3 SAMRS-MAE

We conduct MAE pre-training [8] on the SAMRS training set. To improve the pre-training performance, we further clip the image to $384 \times 384$ with a stride of 300, obtaining 609,707 images.

## B.4 Evaluations on the SAMRS Validation Set

Table S3 lists the evaluation results of the pre-trained models on the SAMRS validation sets. For convenience, We uniformly use the images in the size of $512 \times 512$ obtained through a center cropping on SAMRS validation set samples for evaluation. Here, mIOU is adopted as the metric. It can be seen that all scores on the FAST validation set are relatively low, indicating the challenging nature of the proposed dataset. By comparing Table S3 to the fine-tuning results (Table 3-4 in the main text), we can find that model performance on validation and fine-tuning show similar trends. For example, the performances of adopting SEP alone are not as well as with a good initialation. As can be seen, compared to UNet, UperNet achieves higher accuracies, indicating the model representations are more expressive. Therefore, UperNet can obtain better performances on the challenging IsAID dataset. In addition, it can be observed that the performances of vision transformer networks still surpass convolutional networks, especially for the hierarchical structure. We notice the InternImage performs poorly with $512 \times 512$ images. Therefore, we resize the input image to $224 \times 224$, and the accuracy is recovered. Note the setting of $224 \times 224$ is adopted in pre-training. These results indicate the InternImage may be more dependent on the input size of the pre-training. We have not presented the results of Mask2Former [3] because the validation accuracies on three subsets are close to 0, implying serious over fittings in pre-training since the loss was continuously decreasing. Compared to UNet and UperNet, Mask2former is a newly-proposed framework, it has many different hyperparameters that need to be carefully tuned. Further investigation of hyperparameter settings is required to evaluate its impact on remote sensing images. Nevertheless, we believe the SAMRS validation set is expected to help adjust the settings of SEP in future explorations.

### B.4.1 Dataset

We fine-tune the pre-trained model on two commonly used RS segmentation datasets, including ISPRS Potsdam [1] and iSAID [17]. Before using them, we conduct a series of pre-processing. Here are the details.

**ISPRS Potsdam**: This is the most classical high-resolution RS segmentation dataset. It has 38 large images with an average size of $6,000 \times 6,000$, where the training and testing sets separately include 24 and 14 images. It contains 6 categories: impervious surface, building, low vegetation, tree, car, and clutter. In experiments, we crop the image into $512 \times 512$ with a stride of 320, obtaining 8,664

---

[1]https://www.isprs.org/education/benchmarks/UrbanSemLab/2d-sem-label-potsdam.aspx

Table S3: The mIoUs of different models on SAMRS validation set. †: The input image is resized to 224 × 224. ††: MAE pre-training on the MillionAID. † † †: MAE pre-training on the SAMRS training set

| Method | Pretrain | Backbone | SOTA | SIOR | FAST | Average |
|---|---|---|---|---|---|---|
| UNet [12] | SEP | ResNet-50 | 55.88 | 71.44 | 35.35 | 54.23 |
| UNet | IMP [4]+SEP | ResNet-50 | 58.62 | 80.49 | 37.20 | 58.77 |
| UNet | RSP [14]+SEP | ResNet-50 | 62.43 | 85.37 | 38.41 | 62.07 |
| UperNet [18] | SEP | ResNet-50 | 64.79 | 82.74 | 39.32 | 62.28 |
| UperNet | IMP+SEP | ResNet-50 | 73.59 | 88.59 | 47.10 | 69.76 |
| UperNet | RSP+SEP | ResNet-50 | 76.03 | 90.66 | 47.62 | 71.44 |
| UperNet | IMP+SEP | Swin-T | 81.53 | 94.64 | 57.91 | 78.03 |
| UperNet | IMP+SEP | ViTAEv2-S | 81.13 | 94.06 | 57.90 | 77.70 |
| UperNet | IMP+SEP | InternImage-T | 58.07 | 72.04 | 29.33 | 53.15 |
| UperNet | IMP+SEP | InternImage-T † | 78.06 | 92.51 | 45.70 | 72.09 |
| UperNet | IMP+SEP | ViT-B | 79.37 | 92.76 | 53.05 | 75.06 |
| UperNet | IMP+SEP | ViT-Adapter-B | 80.41 | 93.76 | 51.13 | 75.10 |
| UperNet | BEIT [1] +SEP | ViT-B | 74.10 | 85.81 | 42.04 | 67.31 |
| UperNet | MAE [8] +SEP †† | ViT-B + RVSA | 79.00 | 92.09 | 53.46 | 74.85 |
| UperNet | SAMRS-MAE+SEP † † † | ViT-B + RVSA | 77.64 | 91.87 | 54.78 | 74.67 |

and 5,054 training and testing images. We only use RGB channels. Following most literature in the RS field [14, 20, 6], we ignore the clutter category in training and testing.

**iSAID**: This is a challenging dataset. It provides 15 foregrounds and 1 background category, where 2,806 high-resolution images that range from 800 × 800 to 4,000 × 13,000 pixels are contained. The training, validation, and test sets separately have 1,411, 458, and 937 large images. In this paper, we use the validation set for evaluation since the testing set cannot be acquired. In experiments, we crop the image into 896 × 896 by stride 512, increasing the size of the training and validation set to 33,620 and 11,533. In addition, only the foreground categories are considered.

## C  Datasheet

### C.1  Motivation

**1. For what purpose was the dataset created? Was there a specific task in mind? Was there a specific gap that needed to be filled? Please provide a description.**

**A1:** SAMRS is created to facilitate research in the area of remote sensing (RS) segmentation. Due to the complexity of labeling pixels in remote sensing images (RSIs), the existing RS segmentation field still lacks large-scale RS segmentation datasets, affecting the implementation of RS segmentation pre-training, while SAMRS, a large-scale RS segmentation dataset whose capacity beyond ten times of existing high-resolution RS segmentation datasets, is expected to fill this gap.

**2. Who created this dataset (e.g., which team, research group) and on behalf of which entity (e.g., company, institution, organization)?**

**A2:** SAMRS is created by the first author Di Wang from Wuhan University.

**3. Who funded the creation of the dataset? If there is an associated grant, please provide the name of the grantor and the grant name and number.**

**A3:** The creation of the dataset is founded by the National Natural Science Foundation of China under Grant 62225113 and the National Key Research and Development Program of China under Grant 2022YFB3903405.

### C.2  Composition

**1. What do the instances that comprise the dataset represent (e.g., documents, photos, people, countries)? Are there multiple types of instances(e.g., movies, users, and ratings; people and interactions between them; nodes and edges)? Please provide a description.**

**A1:** SAMRS is comprised of three subsets: SOTA, SIOR, and FAST, which have 18, 20, and 37 categories, respectively. In these sets, each sample consists of a single RSI and corresponding pixel-level semantic labels.

**2. How many instances are there in total (of each type, if appropriate)?**

**A2:** SAMRS has 105,090 images, totally containing 1,668,241 instances.

**3. Does the dataset contain all possible instances or is it a sample (not necessarily random) of instances from a larger set? If the dataset is a sample, then what is the larger set? Is the sample representative of the larger set (e.g., geographic coverage)? If so, please describe how this representativeness was validated/verified. If it is not representative of the larger set, please describe why not (e.g., to cover a more diverse range of instances, because instances were withheld or unavailable).**

**A3:** SAMRS is a real-world sample of global ground objects, including information about their pixel-level category annotations. It is the largest dataset in the area of high-resolution RS segmentation, *e.g.*, 10× more sizes compared with other previous high-resolution RS segmentation datasets. Due to the diversity of real-world ground objects, it is impossible to cover all instances of ground objects on a single dataset. The SAMRS dataset contains three sets that separately have 18, 20, and 37 categories, providing a more diverse set of instances than ever before and will facilitate further studies of RS segmentation.

**4. What data does each instance consist of? "Raw" data (e.g., unprocessed text or images) or features? In either case, please provide a description.**

**A4:** Each instance consists of one land object with its pixel-level semantic annotations and the unprocessed image data.

**5. Is there a label or target associated with each instance? If so, please provide a description.**

**A5:** Yes. Each target is associated with pixel-level semantic labels lying in the corresponding *.png image.

**6. Is any information missing from individual instances? If so, please provide a description, explaining why this information is missing (e.g., because it was unavailable). This does not include intentionally removed information, but might include, e.g., redacted text.**

**A6:** Yes. A limited number of instances may exhibit incomplete masks, as the labels are obtained by the segment anything model (SAM), which cannot ensure perfect segmentation for every object.

**7. Are relationships between individual instances made explicit (e.g., users' movie ratings, social network links)? If so, please describe how these relationships are made explicit.**

**A7:** Yes. The instances' information is stored in the *.png images, and different instances can be clearly by pixel positions and filenames.

**8. Are there recommended data splits (e.g., training, development/validation, testing)? If so, please provide a description of these splits, explaining the rationale behind them.**

**A8:** Yes. We randomly split each subset into train and validation sets following the ratio of 0.95:0.05. The validation set is used for monitoring the training process.

**9. Are there any errors, sources of noise, or redundancies in the dataset? If so, please provide a description.**

**A9:** Since SAMRS is transformed from RS object detection datasets, it may contain noisy labels if the original bounding boxes have errors.

**10. Is the dataset self-contained, or does it link to or otherwise rely on external resources (e.g., websites, tweets, other datasets)? If it links to or relies on external resources, a) are there guarantees that they will exist, and remain constant, over time; b) are there official archival versions of the complete dataset (i.e., including the external resources as they existed at the time the dataset was created); c) are there any restrictions (e.g., licenses, fees) associated with any of the external resources that might apply to a future user? Please provide descriptions of all external resources and any restrictions associated with them, as well as links or other access points, as appropriate.**

**A10:** The SAMRS is comprised of publicly available datasets, including DOTA-V2.0 [5], DIOR [10], and FAIR1M-2.0 [13]. DOTA and FAIR1M can be used for academic purposes only, and FAIR1M uses the Creative Commons Attribution-NonCommercial-ShareAlike 3.0 License. These datasets are publicly available and can be downloaded from their websites. We appreciate the significant contribution of the authors to the research community. Table S4 shows the details that each dataset contributes to the SAMRS dataset.

Table S4: The source datasets adopted in SAMRS.

| Source Dataset | Number of Images | Usage of Original Datasets |
| --- | --- | --- |
| DOTA-V2.0 [5] | 17,480 | Training and validation sets, totally 2,423 images |
| DIOR [10] | 23,463 | Training, validation and testing sets, totally 23,463 images |
| FAIR1M-2.0 [13] | 64,147 | Training and validation sets, totally 24,769 images |

**11. Does the dataset contain data that might be considered confidential (e.g., data that is protected by legal privilege or by doctorpatient confidentiality, data that includes the content of individuals non-public communications)? If so, please provide a description.**

**A11:** No.

**12. Does the dataset contain data that, if viewed directly, might be offensive, insulting, threatening, or might otherwise cause anxiety? If so, please describe why.**

**A12:** No.

### C.3 Collection Process

**1. How was the data associated with each instance acquired? Was the data directly observable (e.g., raw text, movie ratings), reported by subjects (e.g., survey responses), or indirectly inferred/derived from other data (e.g., part-of-speech tags, model-based guesses for age or language)? If data was reported by subjects or indirectly inferred/derived from other data, was the data validated/verified? If so, please describe how.**

**A1:** The data associated with each instance are directly observable, as they are stored in the common png format and can be easily viewed via Python Imaging Library or Open Source Computer Vision Library.

**2. What mechanisms or procedures were used to collect the data (e.g., hardware apparatus or sensor, manual human curation, software program, software API)? How were these mechanisms or procedures validated?**

**A2:** The images in SAMRS come from dataset publicly available datasets described above, which can be directly downloaded from their websites.

**3. If the dataset is a sample from a larger set, what was the sampling strategy (e.g., deterministic, probabilistic with specific sampling probabilities)?**

**A3:** No.

**4. Who was involved in the data collection process (e.g., students, crowdworkers, contractors) and how were they compensated (e.g., how much were crowdworkers paid)?**

**A4:** The first author of this paper.

**5. Over what timeframe was the data collected? Does this timeframe match the creation timeframe of the data associated with the instances (e.g., recent crawl of old news articles)? If not, please describe the timeframe in which the data associated with the instances was created.**

**A5**: Collecting data costs about 1 day, while it takes about 7 days for annotating. Each subset needs to be independently processed by programming, including clipping, unifying filenames and formats, and converting label formats. Finally, we use the SAM to produce labels.

### C.4 Preprocessing/cleaning/labeling

**1. Was any preprocessing/cleaning/labeling of the data done (e.g., discretization or bucketing, tokenization, part-of-speech tagging, SIFT feature extraction, removal of instances, processing**

**of missing values)? If so, please provide a description. If not, you may skip the remainder of the questions in this section.**

**A1:** For the SOTA set, we clip the DOTA-V2.0 images to $1,024 \times 1,024$. For the FAST set, we clip the FAIR1M-2.0 images to $600 \times 600$. Since FAIR1M-2.0 training and validation sets have the same filenames but with different image contents, we rename the FAIR1M-2.0 images and transform the original *.tif files to *.png images. The format of FAIR1M-2.0 annotations is transformed to the format of DOTA-V2.0 for clipping.

In addition, when producing the FAST set, some images of the original FAIR1M-2.0 detection training dataset have wrong position annotations. Hence, we remove these images, as the following files: *1045.tif, 3610.tif, 8693.tif, 14386.tif, 16088.tif, 16385.tif*

**2. Was the "raw" data saved in addition to the preprocessed/cleaned/labeled data (e.g., to support unanticipated future uses)? If so, please provide a link or other access point to the "raw" data.**

**A2:** No.

**3. Is the software used to preprocess/clean/label the instances available? If so, please provide a link or other access point.**

**A3:** We use BboxToolkit for clipping and SAM for labeling.

## C.5   Uses

**1. Has the dataset been used for any tasks already? If so, please provide a description.**

**A1:** No.

**2. Is there a repository that links to any or all papers or systems that use the dataset? If so, please provide a link or other access point.**

**A2:** N/A.

**3. What (other) tasks could the dataset be used for?**

**A3:** SAMRS can be used for the research of RS semantic segmentation pre-training. Besides, it can also be used for specific machine-learning topics such as semantic segmentation, instance segmentation, and object detection. Please see Section 3.1 in the paper.

**4. Is there anything about the composition of the dataset or the way it was collected and preprocessed/cleaned/labeled that might impact future uses? For example, is there anything that a future user might need to know to avoid uses that could result in unfair treatment of individuals or groups (e.g., stereotyping, quality of service issues) or other undesirable harms (e.g., financial harms, legal risks) If so, please provide a description. Is there anything a future user could do to mitigate these undesirable harms?**

**A4:** No.

**5. Are there tasks for which the dataset should not be used? If so, please provide a description.**

**A5:** No.

## C.6   Distribution

**1. Will the dataset be distributed to third parties outside of the entity (e.g., company, institution, organization) on behalf of which the dataset was created? If so, please provide a description.**

**A1:** Yes. The dataset will be publicly available.

**2. How will the dataset will be distributed (e.g., tarball on website, API, GitHub)? Does the dataset have a digital object identifier (DOI)?**

**A2:** It will be publicly available on the project website at GitHub.

**3. When will the dataset be distributed?**

**A3:** The dataset will be distributed once the paper is accepted after peer review.

**4. Will the dataset be distributed under a copyright or other intellectual property (IP) license, and/or under applicable terms of use (ToU)? If so, please describe this license and/or ToU, and provide a link or other access point to, or otherwise reproduce, any relevant licensing terms or ToU, as well as any fees associated with these restrictions.**

**A4:** It will be distributed under the Creative Commons Attribution-NonCommercial-ShareAlike 3.0 License.

**5. Have any third parties imposed IP-based or other restrictions on the data associated with the instances? If so, please describe these restrictions, and provide a link or other access point to, or otherwise reproduce, any relevant licensing terms, as well as any fees associated with these restrictions.**

**A5:** No.

**6. Do any export controls or other regulatory restrictions apply to the dataset or to individual instances? If so, please describe these restrictions, and provide a link or other access point to, or otherwise reproduce, any supporting documentation.**

**A6:** No.

### C.7 Maintenance

**1. Who will be supporting/hosting/maintaining the dataset?**

**A1:** The authors.

**2. How can the owner/curator/manager of the dataset be contacted (e.g., email address)?**

**A2:** They can be contacted via email available on the project website.

**3. Is there an erratum? If so, please provide a link or other access point.**

**A3:** No.

**4. Will the dataset be updated (e.g., to correct labeling errors, add new instances, delete instances)? If so, please describe how often, by whom, and how updates will be communicated to users (e.g., mailing list, GitHub)?**

**A4:** No.

**5. Will older versions of the dataset continue to be supported/hosted/maintained? If so, please describe how. If not, please describe how its obsolescence will be communicated to users.**

**A5:** N/A.

**6. If others want to extend/augment/build on/contribute to the dataset, is there a mechanism for them to do so? If so, please provide a description. Will these contributions be validated/verified? If so, please describe how. If not, why not? Is there a process for communicating/distributing these contributions to other users? If so, please provide a description.**

**A6:** N/A.

## D Visualization

We present more samples of different sets, as shown in Figure S1-S3