# OpenReview forum: "SAMRS: Scaling-up Remote Sensing Segmentation Dataset with Segment Anything Model"
_NeurIPS.cc/2023/Track/Datasets_and_Benchmarks — NeurIPS 2023 Datasets and Benchmarks Poster_

### Official Review · Reviewer_55Yp · 2023-07-20
**Review for paper submission 248**

**Rating:** 6
**Confidence:** 5

**Strengths:**

The approach automates a lot of manual work and translates existing RS datasets for e.g. object detection to segmentation, which enlarges this dataset category. It makes use of very recent approaches in computer vision (i.e. SAM) to benefit the remote sensing field.

Overall a nice approach to accelerate labeling, which is of great interest in the field of RS for many specific use cases (but less for pretraining!) at the moment - rendering this an important topic.

**Additional Feedback:**

Please clarify on the final purpose of this dataset. For pretraining, we see very promising results based on self-supervised MAE. The work should compare to this (e.g.  Wang, D., Zhang, Q., Xu, Y., Zhang, J., Du, B., Tao, D., & Zhang, L. (2022). Advancing plain vision transformer toward remote sensing foundation model. IEEE Transactions on Geoscience and Remote Sensing, 61, 1-15.) and more broadly discuss alternative approaches to supervised pretraining.

**Clarity:**

Besides of the points mentioned under “Opportunities for Improvement”, the paper is well written.

**Correctness:**

Besides of the limitations above, the dataset creation looks scientifically sound to me.

**Documentation:**

As the dataset builds on prior datasets it would be good to mention their licenses, limitations etc.

**Ethics:**

No ethical concerns (as all datasets were already utilized by the community)

**Limitations:**

- Authors state that SAM is not working well on a range of classes in RSI, but just selected ones. This is in line with other literature in the field. However, when utilizing SAM to create data, this can result in datasets that are focusing on classes / images that already work well with existing approaches like SAM, making it difficult to identify blind spots of existing architectures in order to push SOTA in RS. I.e., is this dataset biased towards classes that are already working well with approaches like SAM?

- Inconstancies, errors etc of other datasets used to create this dataset may translate into this new dataset.

- It is not a really novel finding that performance improves with more available data in pretraining, as more data allows to train bigger models with more parameters. See all work going on on self-supervised, transformer-based MAE in RS. Please clarify on the final purpose of this dataset. For pretraining, we see very promising results based on self-supervised MAE. The work should compare to this (e.g.  Wang, D., Zhang, Q., Xu, Y., Zhang, J., Du, B., Tao, D., & Zhang, L. (2022). Advancing plain vision transformer toward remote sensing foundation model. IEEE Transactions on Geoscience and Remote Sensing, 61, 1-15.) and more broadly discuss alternative approaches to supervised pretraining.

**Opportunities For Improvement:**

- The dataset limits former datasets designed for e.g. object detection to segmentation, which was not the original purpose of these datasets. This should be discussed.

- To me it was unclear during dataset creation why all data was used from DIOR dataset, while only training and validation data was used for DOTA and FAIR.

- Instance distribution is what matters most (as all pixels should be required to understand the concept of an instance) - this distribution is heavily imbalanced. This is in contrast to successful datasets for everyday images like ImageNet, which needs clarification.

**Relation To Prior Work:**

The difference from prior work (i.e. work before SAM) is clear.

**Summary And Contributions:**

This paper proposes a new approach to generate masks for semantic segmentation on remote sensing imagery. The authors leverage SAM and existing datasets to create a novel dataset. The contribution is represented by both the dataset and findings on using the proposed dataset in pretraining.

---

> ### Author Response · Authors · 2023-08-26
> **# Response to Reviewer 55Yp (part1: Q1-Q3)**
>
> We sincerely thank you for the careful and thoughtful comments. Below we address the key concerns.
>
> __Q1. The dataset limits former datasets designed for e.g. object detection to segmentation, which was not the original purpose of these datasets. This should be discussed.__
>
> __A1__: Thanks for your question. In fact, object detection and segmentation labels can exist together. For example, the IsAID segmentation dataset [A] and DOTA detection dataset [B] share the same images but with different formats of labels (note the IsAID is independently labeled). Furthermore, some datasets simultaneously have corresponding bounding box annotations and segmentation labels, such as the HRSC2016 testing set [C] and COCO [D]. Therefore, it is reasonable to transform the object detection dataset to a segmentation dataset, especially for object segmentation.
>
> In the revised manuscript, we discuss this issue in Section 4.4:
>
> *Previous classical datasets, such as HRSC2016 [22] and COCO [19], simultaneously contain bounding box and pixel-level mask annotations, proving the feasibility of the coexistence for segmentation and detection labels. Therefore, it is reasonable to construct the SAMRS dataset by transforming existing RS object detection datasets...*
>
> [A] Syed Waqas Zamir, Aditya Arora, Akshita Gupta, Salman Khan, Guolei Sun, Fahad Shahbaz Khan, Fan Zhu, Ling Shao, Gui-Song Xia, and Xiang Bai. isaid: A large-scale dataset for instance segmentation in aerial images. In Proceedings of the IEEE Conference on Computer Vision and Pattern Recognition Workshops, pages 28–37, 2019.
>
> [B] Gui-Song Xia, Xiang Bai, Jian Ding, Zhen Zhu, Serge Belongie, Jiebo Luo, Mihai Datcu, Marcello Pelillo, and Liangpei Zhang. Dota: A large-scale dataset for object detection in aerial images. In The IEEE Conference on Computer Vision and Pattern Recognition (CVPR), June 2018.
>
> [C] Zikun Liu, Liu Yuan, Lubin Weng, and Yiping Yang. A high resolution optical satellite image dataset for ship recognition and some new baselines. In ICPRAM, pages 324–331, 2017.
>
> [D] Tsung-Yi Lin, Michael Maire, Serge Belongie, James Hays, Pietro Perona, Deva Ramanan, Piotr Dollár, and C Lawrence Zitnick. Microsoft coco: Common objects in context. In Proceedings of the European conference on computer vision (ECCV), pages 740–755, 2014.
>
> __Q2. To me it was unclear during dataset creation why all data was used from DIOR dataset, while only training and validation data was used for DOTA and FAIR.__
>
> __A2__: Thanks for your question. For DOTA and FAIR1M, we only transform train and validation sets because their testing sets are private and only used to evaluate detection methods online. For DIOR, since the labels of all data are public, we transform all samples.
>
> __Q3. Instance distribution is what matters most (as all pixels should be required to understand the concept of an instance) - this distribution is heavily imbalanced. This is in contrast to successful datasets for everyday images like ImageNet, which needs clarification.__
>
> __A3__: Thanks for your question. Different from the classification dataset, e.g., ImageNet, segmentation datasets are pixel-wise. Therefore, the category imbalance of segmentation datasets depends on the sizes and distributions of the objects in the real world. The classes in existing segmentation datasets are also imbalanced, such as the Cityscapes dataset.
>
> In Figure 5, we have separately displayed the pixel-level (abc) and instance-level (def) category distributions of SAMRS subsets: SOTA, SIOR, and FAST. Although the categories are imbalanced, it can be seen that compared to the instance-level distribution, the pixel-level distribution is more moderate. The pixel-level distribution is more important for the semantic segmentation task.
>
> In fact, the heavily imbalanced instance-level distributions cannot fully reflect pixel-level distributions, such as the football field in the FAST set. We have analyzed these distributions in paragraph 1 of Section 3.2.
>
>
> *...In the instance-level statistics, we observe a large number of vehicle annotations, particularly on small ships and cars, as they are common in the real world and frequently appear in RSIs. This could also be the goal of initially developing these detection datasets. For instance, DOTA-V2.0 focuses on small targets, while FAIR1M mainly aims to accurately distinguish between different types of vehicles. **Furthermore, it is observed that some categories have a high number of pixels but a low number of instances, which is likely due to their large size. For instance, the expressway-service-area in SIOR and the football-field in FAST demonstrate this pattern.***

---

> ### Author Response · Authors · 2023-08-26
> **# Response to Reviewer 55Yp (part2: Q4-Q5)**
>
> __Q4. Authors state that SAM is not working well on a range of classes in RSI, but just selected ones. This is in line with other literature in the field. However, when utilizing SAM to create data, this can result in datasets that are focusing on classes / images that already work well with existing approaches like SAM, making it difficult to identify blind spots of existing architectures in order to push SOTA in RS. I.e., is this dataset biased towards classes that are already working well with approaches like SAM?__
>
> __A4__: Thanks for your question. We have not identified that SAM does not work well on a range of classes in RSI. On the contrary, we claim it performs well on RSI, as shown in paragraph 3 of Section 1:
>
> *...We have also found it performs well in recognizing diverse targets in RSI, even when the images are obtained using sensors that perceive different bands, such as infrared and microwave, or with varying resolutions, such as airborne or satellite imagery, as illustrated in Figure 1. Although we acknowledge that SAM may not have fully detected all regions, we believe that it has significant potential to improve the efficiency of annotating RSIs since it delivers promising segmentations on recognized areas.*
>
> In fact, the recognition of SAM is class-agnostic, i.e., it is not affected by the target category. It can describe the contours of any objects and produce corresponding masks if given suitable prompts. Therefore, the generation of SAMRS will not be affected by the preferences of SAM. In SAMRS, the obtained categories are determined by the used object detection dataset, while the qualities of the generated masks are more related to the employed prompts. We have determined the optimal setting of prompts by experiments, see Table 1.
>
> __Q5. Inconstancies, errors etc of other datasets used to create this dataset may translate into this new dataset.__
>
> __A5__: Thanks for your question. To avoid such problems, we check the object detection dataset and remove some samples whose bounding boxes have large positional errors, as shown in the datasheet-C.4-A1 of supplementary material. We must acknowledge that there are some small errors in the original detection dataset that may transfer to the obtained segmentation dataset. However, it is noteworthy that existing large-scale datasets may contain label noise, such as ImageNet, but it is enough for supervised pre-training and learning discriminative feature representation. We follow the same idea and use the proposed SAMRS dataset for pre-training.

---

> ### Author Response · Authors · 2023-08-26
> **# Response to Reviewer 55Yp (part3: Q6 part1)**
>
> __Q6. It is not a really novel finding that performance improves with more available data in pre-training, as more data allows to train bigger models with more parameters. See all work going on self-supervised, transformer-based MAE in RS. Please clarify on the final purpose of this dataset. For pre-training, we see very promising results based on self-supervised MAE. The work should compare to this (e.g. Wang, D., Zhang, Q., Xu, Y., Zhang, J., Du, B., Tao, D., \& Zhang, L. (2022). Advancing plain vision transformer toward remote sensing foundation model. IEEE Transactions on Geoscience and Remote Sensing, 61, 1-15.) and more broadly discuss alternative approaches to supervised pre-training.__
>
> __A6__: Thanks for your question. Here, we must clarify that the SAMRS is used to tackle the task-level discrepancy between pre-training and downstream segmentation tasks instead of initializing weights as the usual pre-training. Therefore, the SEP on SAMRS must be implemented based on existing pretrained weights, as described in the revised manuscript in paragraph 1 of Section 4.1.2:
>
> *Intuitively, a well-initialized backbone network generates discriminative features at the beginning of training, thereby facilitating the optimization process of the decoder component.  Since the SEP is expected to mitigate the gaps between pre-training and downstream tasks. To this end, before the segmentation pre-training phase, the selected model's backbone network is initialized using pretrained weights. In our experiments, to fully evaluate the SEP, besides basic supervised pre-training on ImageNet (IMP) [7], we also utilize the RSP [36] on the MillionAID Dataset [23]. In addition, the unsupervised pre-training weights are also involved, including BEiT [1] and MAE [13]. Here, the MAE pre-training is conducted on the MillionAID [37], while the BEiT is pretrained on the ImageNet.*
>
> Nevertheless, in the revised manuscript, we follow your suggestion and use SEP alone for pre-training. The results are shown in Table 3 (related part). As can be seen, adopting SEP alone obtains comparable performance as using IMP but achieves inferior performance than using IMP and SEP together since it lacks a good initialization.
>
> *Table 3. Segmentation results of different methods on the ISPRS Potsdam dataset.*
>
> | Method | Pretrain | Backbone | OA | mF1 |
> | :------ | :-- | :---: | :---: | :---: |
> | UNet | SEP | ResNet-50 | 89.70 | 90.18 |
> | UNet | IMP | ResNet-50 | 89.94 | 90.47 |
> | UNet | IMP+SEP | ResNet-50 | 90.29 | 90.82 |
> | UperNet | SEP | ResNet-50 | 89.70 | 90.21 |
> | UperNet | IMP | ResNet-50 | 89.59 | 90.07 |
> | UperNet | IMP+SEP | ResNet-50 | 90.27 | 90.76 |
>
> Correspondingly, Section 4.2.1 is revised to:
>
> *...It can be seen that, without good initialization, the performances of SEP are comparable as IMP but inferior to IMP+SEP...*
>
> To further prove this, in the revised manuscript, we additionally conduct the SEP based on the weights generated by implementing the MAE on the SAMRS training set, see Table 3 (related part).
>
> *Table 3. Segmentation results of different methods on the ISPRS Potsdam dataset. $\ddagger\ddagger$: MAE pre-training on the SAMRS training set.*
>
> | Method | Pretrain | Backbone | OA | mF1 |
> | :------ | :-- | :---: | :---: | :---: |
> | UperNet | SAMRS-MAE $\ddagger\ddagger$ | ViT-B + RVSA |  90.71 | 90.13 |
> | UperNet | SAMRS-MAE+SEP | ViT-B + RVSA | 91.01 | 91.43 |
>
> It can be seen that the SEP still improves the network with the initialized weights pretrained on the SAMRS training set. These results indicate that the effectiveness of SEP is not from pre-training with more available data. It surely bridges the gaps between pre-training and downstream segmentation tasks.

---

> ### Author Response · Authors · 2023-08-26
> **# Response to Reviewer 55Yp (part4: Q6 part2-Q7)**
>
> To completely show the effectiveness of SAMRS and SEP, in the revised manuscript, we adopt more diverse backbones (ResNet, Swin, ViTAEv2, InternImage, ViT, ViT-Adapter, and ViT-RVSA), decoders (UNet, UperNet), and pretrained weights (IMP, RSP, BEiT, MAE, SAMRS-MAE), as shown in Table 3-4. The related analyses are presented in Section 4.2.1:
>
> *In the RS community, ISPRS Potsdam and IsAID are commonly-used finely annotated datasets for evaluating segmentation methods [25,28,36,37,44], and we use them to assess the pre-trained models, as Table 3-4 shown. It can be seen that, without good initialization, the performances of SEP are comparable as IMP but inferior to IMP+SEP. On the Potsdam scene, for the traditional encoder-decoder network, SEP improves both convolutional and vision transformer networks, especially for UperNet and the backbones containing hierarchical features (also includes ResNet). As a result, ViTAEv2-S is greatly boosted and overperforms existing advanced methods in overall accuracy. We also observe that SEP is useful when combined with different pre-training strategies. Even if using the initialized weights generated by pre-training on SAMRS itself, SEP still can improve the accuracy, excluding the effect of data volume used for training. We notice SEP played a negative role in the end-to-end structure, it may be because the objects of SAMRS are too small, which is unfavorable for the region-based Mask2Former. In addition, the Mask2Former, which has obtained high accuracies on natural images, does not perform as well as UNet and UperNet on RSIs. These results indicate more refined parameter adjustments of Mask2Former are needed in later research. On the IsAID dataset, the performances of SEP on simple convolutional networks depending on local perception are unstable, because IsAID and DOTA share the same images but with different annotations, which may confuse the model. Benefiting from SEP, vision transformer networks are further enhanced and surpass previous methods. From these results we can see, SEP is able to mitigate the influence of task-level disparities, specifically the gaps between upstream pre-training tasks and downstream segmentation tasks.*
>
>
> In summary, SEP can be compatible with and improve a variety of models and pre-training strategies.
>
> What's more, in the revised manuscript, we follow the reviewer's suggestion and use the ViT-RVSA (reviewer-mentioned paper) to evaluate SEP on the Potsdam dataset, see Table 3:
>
> *Table 3. Segmentation results of different methods on the ISPRS Potsdam dataset. $\ddagger$: MAE pre-training on the MillionAID. $\ddagger\ddagger$: MAE pre-training on the SAMRS training set.*
>
> | Method | Pretrain | Backbone | OA | mF1 |
> | :------ | :-- | :--- | :---: | :---: |
> | UperNet | MAE $\ddagger$ | ViT-B + RVSA |  90.97 | 90.37 |
> | UperNet | MAE+SEP | ViT-B + RVSA | 91.33 |  91.69 |
> | UperNet | SAMRS-MAE $\ddagger\ddagger$ | ViT-B + RVSA | 90.71 | 90.13 |
> | UperNet | SAMRS-MAE+SEP | ViT-B + RVSA | 91.01 | 91.43 |
>
> In addition, we compare the performances between SEP pretrained models and ViTAE-B + RVSA$\Diamond$ on the IsAID, as shown in Table 4:
>
> *Table 4. Segmentation results of different methods on the IsAID dataset. $\ddagger$: MAE pre-training on the MillionAID.*
>
> | Method | Pretrain | Backbone  | mIOU |
> | :------ | :-- | :--- | :---: |
> | HMANet [28] | IMP | ResNet-50 | 62.64 |
> | UperNet | MAE $\ddagger$ | ViTAE-B + RVSA$^\Diamond$ | 64.49 |
> | FactSeg [25] | IMP | ResNet-50 |  64.79 |
> | RssFormer [44] | IMP | RSS-B |  65.88 |
> | UNet | IMP | ResNet-50 |35.80 |
> | UNet | IMP+SEP | ResNet-50 |  54.00 |
> | UNet | RSP | ResNet-50 |  46.32 |
> | UNet | RSP+SEP | ResNet-50 | 30.45 |
> | UperNet | IMP | ResNet-50 |  62.54 |
> | UperNet | IMP+SEP | ResNet-50 | 58.10 |
> | UperNet | RSP | ResNet-50 | 32.97 |
> | UperNet | RSP+SEP | ResNet-50 | 59.26 |
> | UperNet | IMP | Swin-T |  63.77 |
> | UperNet | IMP+SEP | Swin-T |  64.25 |
> | UperNet | IMP | ViTAEv2-S | 65.79 |
> | UperNet | IMP+SEP | ViTAEv2-S |  66.26 |
> | UperNet | IMP | InternImage-T |  65.02 |
> | UperNet | IMP+SEP | InternImage-T | 66.05 |
>
> __Q7. As the dataset builds on prior datasets it would be good to mention their licenses, limitations etc.__
>
> __A7__: Thanks for your suggestion. In the revised manuscript, we add the licenses of the used DOTA, DIOR, and FAIR1M datasets in Section C.2-A10 of supplementary material. Since FAIR1M uses the Creative Commons Attribution-NonCommercial-ShareAlike 3.0 License, we distribute the SAMRS using the same license, and Section C.6-A4 is correspondingly revised.
>
> In addition, in the revised manuscript, we also explain their licenses in Section 2.2.

---

> ### Author Response · Authors · 2023-08-29
> **Request for Discussion**
>
> Dear Reviewer 55Yp
>
> We sincerely thank you again for your great efforts in reviewing this paper. We have addressed your major concerns about the rationality of creating segmentation datasets by transforming existing remote sensing object detection datasets using the SAM, the characteristics of the obtained datasets, and the final purpose of the SAMRS. Please don't hesitate to let us know if there are still concerns/questions. Your insights are important and valuable to us.
>
> Kind regards,
>
> Submission 248 Authors

---

> ### Author Response · Authors · 2023-08-31
> **Thanks for your reviews**
>
> Dear Reviewer 55Yp
>
> The discussion period is approaching. We have provided detailed responses and supplemented more experimental results. We hope they can address your concerns. Thanks for your valuable comments for improving the quality of our study. If you have other concerns, please do not hesitate to contact us, we are glad to resolve them.
>
> Kind regards,
>
> Submission 248 Authors

---

> > ### Comment · Reviewer_55Yp · 2023-08-31
> >
> > Dear authors,
> >
> > thank you for your explanations. The work has meaningfully improved based on the additional experiments, I am happy to improve my score to 6.
> >
> > Best regards

---

### Official Review · Reviewer_rfwi · 2023-07-21
**SAMRS is a valuable new RS dataset that is well-motivated and thoughtfully constructed, but more discussion about label quality is needed.**

**Rating:** 8
**Confidence:** 4
**Clarity:** The paper is well-written and easy to…

**Strengths:**

- The authors carefully design methods to automatically "translate" the existing dataset labels to detailed segmentation labels by developing careful prompts for SAM. They describe the methodology in detail with clarity and quantitatively assess different labeling methods.
- They provide useful quantitative analysis of the resulting labeled dataset to better understand the automatically labeled segmentation masks included in SAMRS.
- They conduct several interesting pre-training experiments, comparing to image classification pre-training and evaluating in fine-tuning settings with a small amount of data, which help shed light on the utility of SAMRS.
- Overall the paper is well-written, and both the methodology and described dataset are likely of interest to the broader research community. The methodology is general enough to be applied to other datasets, potentially beyond remote sensing, and the dataset will be a valuable resource to both the remote sensing and ML communities for improving ML methods on remote sensing data.
- Ethical and social implications are minimal as the authors have mostly just (substantially) improved existing datasets.
- The authors also provide a datasheet which includes a lot of useful information for users of the dataset.

**Additional Feedback:**

- "To the best of our knowledge, it is the first time that SAM has been employed for labeling RSIs". Several recent works have been exploring SAM for remote sensing. I encourage the authors to verify the validity of this statement and rephrase if necessary (and cite any appropriate + relevant recent works).
- Line 161:  "10,5090" the comma is in the wrong place
- The total number of labeled object instances are reported in the datasheet, but I think this number belongs in the main text, and potentially the abstract.
- Is it possible to show the text labels somehow in Figure 4? That would help with interpreting the labels.

Overall I think the authors have done a great job preparing a highly valuable resource for the community.

**Correctness:**

To the best of my knowledge, the claims in the submission are correct and the dataset is constructed in a sound way.

**Documentation:**

- Data collection is highly detailed.
- Format of the labels could be explained better.
- Availability is clear.
- Maintenance is explained in the datasheet.
- Nothing is stated about ethical and responsible use.
- The license is stated in the datasheet.
- There seems to be sufficient detail to support reproducibility of the benchmark experiments.

**Ethics:**

Not that I can think of. If the authors haven't already, I would make sure the included datasets have licenses that allow for redistribution.

**Limitations:**

The authors include a brief limitations section, but I think it adequately addresses the limitations and potential negative societal impact of their work, which I believe are minimal.

**Opportunities For Improvement:**

- Given the dataset was labeled automatically, there should be more discussion about label quality. The original SAM paper addresses this by keeping a human-in-the-loop, but it seems SAMRS does not do this (understandably as this is resource intensive). The authors do share some quantitative results in Table 1, but the test set is quite small and it's not clear how these results apply to the full dataset. Including more expansive qualitative analysis at the very least would be beneficial (e.g. expanding in section 3.3.).
- The choice of the four datasets included should be motivated. Why were these datasets chosen?
- The choice of using Potsdam for evaluating the fine-tuning approaches should be motivated.
- It seems like the authors only evaluate a staged pre-training strategy, where SEP is done after either ImageNet pre-training or MillionAID pre-training. Why not evaluate doing SEP on SAMRS alone, without doing other pre-training first?

**Relation To Prior Work:**

This is clearly discussed.

**Summary And Contributions:**

The authors use the recently developed Segment Anything Model (SAM) to produce new segmentation labels for four publicly available remote sensing datasets, resulting in an aggregated dataset called SAMRS, which is larger than all previous high-resolution remote sensing datasets for segmentation. They analyze the SAMRS labels and conduct pre-training experiments with it to benchmark it against other pre-training strategies/datasets.

---

> ### Author Response · Authors · 2023-08-26
> **# Response to Reviewer rfwi (part1: Q1)**
>
> We sincerely thank you for the careful and thoughtful comments. Below we address the key concerns.
>
> __Q1: Given the dataset was labeled automatically, there should be more discussion about label quality. The original SAM paper addresses this by keeping a human-in-the-loop, but it seems SAMRS does not do this (understandably as this is resource intensive). The authors do share some quantitative results in Table 1, but the test set is quite small and it's not clear how these results apply to the full dataset. Including more expansive qualitative analysis at the very least would be beneficial (e.g. expanding in section 3.3.).__
>
> __A1__: Thanks for your comment. The HRSC2016 detection set is used to evaluate the label quality since 124 images of them provide both detection and segmentation labels, as shown in Section 2.2:
>
> *...**Additionally, in the testing set, 124 images possess bounding box annotations and pixel-level labels simultaneously, making it highly suitable for evaluating the accuracy of SAM annotations. Therefore, we conduct an ablation study on the testing set consisting of the aforementioned 124 images to determine the optimal configuration for SAM.** Following this, we generate segmentation labels for the remaining datasets...*
>
> Concretely, we consider different prompts, including center point (CP), horizontal bounding box (H-Box), and H-Box-enclosed mask (H-Box-M). Regarding the characteristics of remote sensing images, we extra consider the rotated bounding box (R-Box). However, SAM does not directly support R-Box as the prompt. Therefore, we use the minimum circumscribed horizontal rectangle of the R-Box, denoted by RH-Box. In addition, the mask of R-Box and RH-Box, denoted by R-Box-M and RH-Box-M, can also be employed.
>
> Six basic prompts, including CP, H-Box, RH-Box, H-Box-M, R-Box-M, and RH-Box-M, and their combinations are evaluated on the aforementioned 124 images, as shown in Table 1.
>
> In fact, these 124 images are manually labeled by experts, and it is unnecessary for us to annotate again. With these ground truths, we can do a label quality control. This procedure expresses a thinking of human-in-the-loop, i.e., using true labels to obtain optimal settings, which are then applied to the whole dataset.
>
> According to Table 1, we obtain a conclusion for generating segmentation annotations, as shown in Section 2.4:
>
> *...From this experiment, we conclude that: **if an RS object detection dataset only has R-Box annotations, then the RH-Box prompt should be used; otherwise, the H-Box prompt should be adopted.** This consideration is applied in our later dataset transformations.*
>
> Then we adopt this conclusion on large-scale remote sensing detection datasets: DOTA-V2.0, DIOR, and FAIR1M-2.0 to produce the SAMRS segmentation dataset, as shown in Section 2.5:
>
> *For the FAIR1M-2.0 dataset, since it only contains R-Box annotations, we use the corresponding RH-Box as the prompt. For DOTA-V2.0 and DIOR, we directly adopt the H-Box prompt...*
>
> In summary, the 124 images of the HRSC2016 testing set act as the ground truth for quality control of the produced segmentation labels using the proposed SAM-based pipeline. Besides, we conduct additional label quality control by removing some samples whose bounding boxes have large positional errors, please see the datasheet-C.4-A1.
>
> It is also noteworthy that existing large-scale datasets may contain label noise, such as ImageNet, but it is enough for supervised pre-training and learning discriminative feature representation. We follow the same idea and use the proposed SAMRS dataset for pre-training.

---

> ### Author Response · Authors · 2023-08-26
> **# Response to Reviewer rfwi (part2: Q2-Q3)**
>
> __Q2: The choice of the four datasets included should be motivated. Why were these datasets chosen?__
>
> __A2__: Thanks for your suggestion. The HRSC2016 dataset is used for label quality control, we have explained it in question 1. We choose DOTA, DIOR, and FAIR1M because they are three large-scale remote sensing target detection datasets, as shown in [A].
>
> [A] Xian Sun, Peijin Wang, Zhiyuan Yan, Feng Xu, Ruiping Wang, Wenhui Diao, Jin Chen, Jihao Li, Yingchao Feng, Tao Xu, Martin Weinmann, Stefan Hinz, Cheng Wang, and Kun Fu. Fair1m: A benchmark dataset for fine-grained object recognition in high-resolution remote sensing imagery. ISPRS Journal of Photogrammetry and Remote Sensing, 184:116-130, 2022.
>
> We have added the related motivation in the revised manuscript, see Section 2.2:
>
> *In this study, we employ SAM on four public RS object detection datasets, namely HRSC2016 [22], DOTA-V2.0 [10], DIOR [18], and FAIR1M-2.0 [33]. DOTA, DIOR and FAIR1M are three large-scale datasets [33]. HRSC2016 is primarily designed for ship detection and comprises only one category. In comparison to the other three datasets, it has the smallest data volume. Additionally, in the testing set, 124 images possess bounding box annotations and pixel-level labels simultaneously, making it highly suitable for evaluating the accuracy of SAM annotations. Therefore, we conduct an ablation study on the testing set consisting of the aforementioned 124 images to determine the optimal configuration for SAM. Following this, we generate segmentation labels for the remaining datasets...*
>
> It is expected that the developed method can be applied to other large-scale remote sensing detection datasets to obtain more segmentation labels.
>
> __Q3: The choice of using Potsdam for evaluating the fine-tuning approaches should be motivated.__
>
> __A3__: Thanks for your suggestion. We select frequently used and finely annotated public datasets for finetuning. Besides, we utilize another popular iSAID dataset for finetuning to more comprehensively evaluate the SEP. The related motivation is presented in the new version manuscript, see Section 4.2.1:
>
> *In the RS community, ISPRS Potsdam and IsAID are commonly-used finely annotated datasets for evaluating segmentation methods [25, 28, 36, 37, 44], and we use them to assess the pre-trained models, as Table 3-4 shown...*

---

> ### Author Response · Authors · 2023-08-26
> **# Response to Reviewer rfwi (part3: Q4-Q6)**
>
> __Q4: It seems like the authors only evaluate a staged pre-training strategy, where SEP is done after either ImageNet pre-training or MillionAID pre-training. Why not evaluate doing SEP on SAMRS alone, without doing other pre-training first?__
>
> __A4__: Thanks for your question. We observe the gap between classification and segmentation tasks, as shown in paragraph 5 of Section 1:
>
> *Since we efficiently obtain numerous segmentation label maps, it is natural to consider using the obtained dataset for pre-training. **Existing models pretrained by classification tasks may be not very suitable for downstream tasks, e.g., segmentation, because of the task-level discrepancy [36], while the emergence of SAMRS is expected to address this issue.** To this end, we train classical deep learning models on the SAMRS, and finetune the trained model on typical RS segmentation datasets to explore the feasibility of segmentation pre-training.*
>
> In this paper, the SEP is used to address the task-discrepancy instead of pre-training, thus we combine it with different initialized weights, see paragraph 1 of Section 4.1.2:
>
> *Intuitively, a well-initialized backbone network generates discriminative features at the beginning of training, thereby facilitating the optimization process of the decoder component. Since the SEP is expected to mitigate the gaps between pre-training and downstream tasks. To this end, before the segmentation pre-training phase, the selected model's backbone network is initialized using pretrained weights. In our experiments, to fully evaluate the SEP, besides basic supervised pre-training on ImageNet (IMP) [7], we also utilize the RSP [36] on the MillionAID Dataset [23]. In addition, the unsupervised pre-training weights are also involved, including BEiT [1] and MAE [13]. Here, the MAE pre-training is conducted on the MillionAID [37], while the BEiT is pretrained on the ImageNet.*
>
> Following your suggestion, we also use SEP alone for pre-training. The results are shown in Table 3 (related part). As can be seen, adopting SEP alone obtains comparable performance as using IMP but achieves inferior performance than using IMP and SEP together since it lacks a good initialization.
>
> *Table 3. Segmentation results of different methods on the ISPRS Potsdam dataset.*
>
> | Method | Pretrain | Backbone | OA | mF1 |
> | :------ | :-- | :---: | :---: | :---: |
> | UNet | SEP | ResNet-50 | 89.70 | 90.18 |
> | UNet | IMP | ResNet-50 | 89.94 | 90.47 |
> | UNet | IMP+SEP | ResNet-50 | 90.29 | 90.82 |
> | UperNet | SEP | ResNet-50 |  89.70 | 90.21 |
> | UperNet | IMP | ResNet-50 | 89.59 | 90.07 |
> | UperNet | IMP+SEP | ResNet-50 |  90.27 | 90.76 |
>
> Correspondingly, Section 4.2.1 is revised to:
>
> *...It can be seen that, without good initialization, the performances of SEP are comparable as IMP but inferior to IMP+SEP...*
>
> __Q5: Format of the labels could be explained better.__
>
> __A5__: Thanks for your suggestion. In the revised manuscript, the label format of SAMRS is introduced in Section 3.1:
>
> *The obtained segmentation labels are stored in \*.png files. Pixel values are aligned with the object classess of source object detection datasets. The areas that have not been covered by the generated masks will be in a pixel value of 255. We present the comparison of our SAMRS dataset with existing high-resolution RS segmentation datasets in Table 2 from different aspects...*
>
> __Q6: ``To the best of our knowledge, it is the first time that SAM has been employed for labeling RSIs''. Several recent works have been exploring SAM for remote sensing. I encourage the authors to verify the validity of this statement and rephrase if necessary (and cite any appropriate + relevant recent works)..__
>
> __A6__: Thanks for your comment. In the revised manuscript, we summarize related works that also apply SAM to the RS field in paragraph 3 of Section 1:
>
> *...In the RS field, [31] firstly tests the performance of SAM on six public datasets. [16] extra introduce a domain decoder to improve the performance of SAM on the planetary geological mapping task. Beyond default prompts, [29,45] consider utilizing texts as the prompt by adopting Grounding DINO [20] to obtain boxes that can be employed by SAM. Then, [29] realizes the one-shot segmentation with the help of PerSAM [47], while [45] applies the heatmap obtained from CLIP [30] to further optimize segmentation results. Different from the above methods with manual prompts, [3] design a prompter to adaptively generate prompts for improving the performance of SAM in instance segmentation. In addition, SAM is also used in producing rotated bounding boxes, which is significant for RS oriented object detection.*
>
> We find that, in the above work, the segmentation results generated by SAM serve as an intermediate product (SAM for planetary geological mapping), or they perform a one-shot text-driven segmentation with the help of Grounding DINO. Therefore, we remove this sentence in the revised manuscript.

---

> ### Author Response · Authors · 2023-08-26
> **# Response to Reviewer rfwi (part4: Q7-Q9)**
>
> __Q7: Line 161: ``10,5090'' the comma is in the wrong place.__
>
> __A7__: Thanks for your comment. This typo has been corrected in the revised manuscript.
>
> __Q8: The total number of labeled object instances are reported in the datasheet, but I think this number belongs in the main text, and potentially the abstract.__
>
> __A8__: Thanks for your suggestion. We have updated the related part in the abstract and paragraph 1 of Section 3.1.
>
> __Q9: Is it possible to show the text labels somehow in Figure 4? That would help with interpreting the labels.__
>
> __A9__: Thanks for your suggestion. We have added corresponding legends in the revised manuscript.

---

> > ### Comment · Reviewer_rfwi · 2023-08-28
> >
> > Thank you to the authors for replying to all of my comments and making corresponding changes to the paper. I believe these changes have improved the paper, but I retain my original score which I believe still accurately reflects the quality and impact of the work.

---

> > > ### Author Response · Authors · 2023-08-29
> > > **Thanks to Reviewer rfwi**
> > >
> > > Dear Reviewer rfwi,
> > >
> > > We are glad that we have addressed most of your concerns. We would like to express our sincere appreciation for your efforts in enhancing the quality of our work.
> > >
> > > Kind regards,
> > >
> > > Submission 248 Authors

---

### Official Review · Reviewer_MpE3 · 2023-07-24
**Review of SAMRS: Scaling-up Remote Sensing Segmentation Dataset with Segment Anything Model**

**Rating:** 5
**Confidence:** 5
**Clarity:** The paper is well-written overall.

**Strengths:**

1. Provides a new perspective that SAM also has a great zero-shot segmentation ability.
2. Authors sufficiently compare different SAM prompt settings according to performance on a certain test set.


**Additional Feedback:**

It is hoped that authors could provide relevant experiments to address my confusion.

**Correctness:**

The construction of the dataset is reasonable and considerate. Authors utilize Segment Anything Model (SAM) to build a large-scale remote sensing images dataset

**Documentation:**

The authors provide details on data collection and organization.

**Ethics:**

No ethics issues.

**Limitations:**

The experiment's baseline, ResNet50 + UNet, is relatively simple. To further demonstrate the effectiveness of the proposed pre-training paradigm, the authors should consider using a state-of-the-art model, like Mask2former.

**Opportunities For Improvement:**

1. Apart from ImageNet pretraining, there are also several effective pretraining strategies and models that have been demonstrated for segmentation tasks, including BEiT pretraining [1], ViT-Adapter [2], and InternImage [3]. It would be valuable to provide experimental evidence comparing these approaches to the proposed methods. Do these approaches conflict with the proposed settings, or are they complementary?
2. Actually, several works have adapted SAM to remote sensing images, such as MMRotate-SAM. These should also be discussed.
[1] Bao H, Dong L, Piao S, et al. Beit: Bert pre-training of image transformers[J]. arXiv preprint arXiv:2106.08254, 2021.
[2] Chen Z, Duan Y, Wang W, et al. Vision transformer adapter for dense predictions[J]. arXiv preprint arXiv:2205.08534, 2022.
[3] Wang W, Dai J, Chen Z, et al. Internimage: Exploring large-scale vision foundation models with deformable convolutions[C]//Proceedings of the IEEE/CVF Conference on Computer Vision and Pattern Recognition. 2023: 14408-14419.


**Relation To Prior Work:**

Authors clearly discuss their contributions with related works.

**Summary And Contributions:**

Authors utilized Segment Anything Model (SAM) to build a large-scale remote sensing images dataset, named SAMRS, from four public remote sensing dataset. Additionally, authors discussed different prompt settings in SAM and provide a suggested setting by two mIoU metrics. Finally, SAMRS is used to pretrain the backbone of segmentation model and achieve a higher performance than other two pretrained settings: ImageNet pre-training and another remote sensing pre-training.

---

> ### Author Response · Authors · 2023-08-26
> **# Response to Reviewer MpE3 (part1: Q1-Q2)**
>
> We sincerely thank you for the careful and thoughtful comments. Below we address the key concerns.
>
> __Q1: Apart from ImageNet pre-training, there are also several effective pre-training strategies and models that have been demonstrated for segmentation tasks, including BEiT pre-training [1], ViT-Adapter [2], and InternImage [3]. It would be valuable to provide experimental evidence comparing these approaches to the proposed methods. Do these approaches conflict with the proposed settings, or are they complementary?__
>
> [1] Bao H, Dong L, Piao S, et al. Beit: Bert pre-training of image transformers[J]. arXiv preprint arXiv:2106.08254, 2021.
>
> [2] Chen Z, Duan Y, Wang W, et al. Vision transformer adapter for dense predictions[J]. arXiv preprint arXiv:2205.08534, 2022.
>
> [3] Wang W, Dai J, Chen Z, et al. Internimage: Exploring large-scale vision foundation models with deformable convolutions[C]//Proceedings of the IEEE/CVF Conference on Computer Vision and Pattern Recognition. 2023: 14408-14419.
>
> __A1__: Thanks for your suggestion. In the revised manuscript, these pre-training strategies and models have been used to evaluate the performance of SEP. From the experimental results, we can find that SEP is able to improve all the above models no matter adopting pre-training strategies and backbones. The details results of the ISPRS Potsdam dataset have been shown in Table 3 (related part):
>
> *Table 3. Segmentation results of different methods on the ISPRS Potsdam dataset.*
>
> | Method | Pretrain | Backbone | OA | mF1 |
> | :------ | :-- | :--- | :---: | :---: |
> | UperNet | IMP | InternImage-T | 91.65 | 91.18 |
> | UperNet | IMP+SEP | InternImage-T  |  92.08 |  92.41 |
> | UperNet | IMP | ViT-Adapter-B | 91.53 | 90.98 |
> | UperNet | IMP+SEP | ViT-Adapter-B |  91.91 |  92.24 |
> | UperNet | BEiT | ViT-B |  86.86 | 87.49 |
> | UperNet | BEiT+SEP | ViT-B |   88.20 |  88.76 |
>
>
> We additionally conduct experiments on the iSAID dataset, as shown in Table 4 (related part).
>
> *Table 4. Segmentation results of different methods on the IsAID dataset.*
> | Method | Pretrain | Backbone  | mIOU |
> | :------ | :-- | :---: | :---: |
> | UperNet | IMP | InternImage-T |  65.02 |
> | UperNet | IMP+SEP | InternImage-T | 66.05 |
>
>
> It can be seen that the SEP improves the performance of InternImage on the IsAID dataset.
>
> __Q2: Actually, several works have adapted SAM to remote sensing images, such as MMRotate-SAM. These should also be discussed.__
>
> __A2__: Thanks for your suggestion. In the revised manuscript, we discuss related works that also apply SAM to the RS field in paragraph 3 of Section 1:
>
> *...In the RS field, [31] firstly tests the performance of SAM on six public datasets. [16] extra introduce a domain decoder to improve the performance of SAM on the planetary geological mapping task. Beyond default prompts, [29,45] consider utilizing texts as the prompt by adopting Grounding DINO [20] to obtain boxes that can be employed by SAM. Then, [29] realizes the one-shot segmentation with the help of PerSAM [47], while [45] applies the heatmap obtained from CLIP [30] to further optimize segmentation results. Different from the above methods with manual prompts, [3] design a prompter to adaptively generate prompts for improving the performance of SAM in instance segmentation. In addition, SAM is also used in producing rotated bounding boxes, which is significant for RS oriented object detection.*

---

> ### Author Response · Authors · 2023-08-26
> **# Response to Reviewer MpE3 (part2: Q3)**
>
> __Q3: The experiment's baseline, ResNet50 + UNet, is relatively simple. To further demonstrate the effectiveness of the proposed pre-training paradigm, the authors should consider using a state-of-the-art model, like Mask2former.__
>
> __A3__: Thanks for your suggestion. In the revised manuscript, the mask2former has been employed. However, we observe the accuracy is decreased when adopting SEP, as shown in the related part of Table 3:
>
> *Table 3. Segmentation results of different methods on the ISPRS Potsdam dataset.*
>
> | Method | Pretrain | Backbone | OA | mF1 |
> | :------ | :-- | :---: | :---: | :---: |
> | Mask2Former | IMP | ResNet-50 |  87.54 | 86.87 |
> | Mask2Former | IMP+SEP | ResNet-50 | 70.14 | 69.86 |
>
> In our consideration, there may be the following reasons: although Mask2former is an advanced segmentation framework and has achieved high accuracies on natural images, its performance on RS images is not very well. This finding can be obtained by comparing Mask2Former, UNet and UperNet with a backbone of ResNet-50 in Table 3:
>
> *Table 3. Segmentation results of different methods on the ISPRS Potsdam dataset.*
>
> | Method | Pretrain | Backbone | OA | mF1 |
> | :------ | :-- | :---: | :---: | :---: |
> | UNet | IMP | ResNet-50 | 89.94 | 90.47 |
> | UperNet | IMP | ResNet-50 | 89.59 | 90.07 |
> | Mask2Former | IMP | ResNet-50 |  87.54 | 86.87 |
>
> In addition, many objects in SAMRS are too small, which may be unsuitable for the region-based Mask2Former. Another reason is, compared to UNet and UperNet, Mask2former is a newly-proposed framework, it has many different hyperparameters that need to be carefully tuned. Due to limited time, we only conduct one group experiment. Further investigation of hyperparameter settings is required to evaluate the impact of SEP in the future. Correspondingly, Section 4.2.1 is revised to:
>
> *...We notice SEP played a negative role in the end-to-end structure, it may be because the objects of SAMRS are too small, which is unfavorable for the region-based Mask2Former. In addition, the Mask2Former, which has obtained high accuracies on natural images, does not perform as well as UNet and UperNet on RSIs. These results indicate more refined parameter adjustments of Mask2Former are needed in later research...*
>
> Furthermore, as already mentioned, besides UNet and Mask2Former, we also utilize UperNet, which is widely used for evaluating the performance of different backbones on semantic segmentation tasks. Here we show some examples from Table 3.
>
> *Table 3. Segmentation results of different methods on the ISPRS Potsdam dataset.*
>
> | Method | Pretrain | Backbone | OA | mF1 |
> | :------ | :-- | :---: | :---: | :---: |
> | UperNet | SEP | ResNet-50 |  89.70 | 90.21 |
> | UperNet | IMP | ResNet-50 |  89.59 | 90.07 |
> | UperNet | IMP+SEP | ResNet-50 |  90.27 |  90.76 |
>
> Correspondingly, Figure 7 is also updated.

---

> ### Author Response · Authors · 2023-08-29
> **Request for Discussion**
>
> Dear Reviewer MpE3
>
> We sincerely thank you again for your great efforts in reviewing this paper. We have addressed your major concerns about the discussion about other SAM-based works and the validations under different backbones, pre-training strategies, and segmentation frameworks. Please don't hesitate to let us know if there are still concerns/questions. Your insights are important and valuable to us.
>
> Kind regards,
>
> Submission 248 Authors

---

> ### Author Response · Authors · 2023-08-31
> **Thanks for your reviews**
>
> Dear Reviewer MpE3
>
> The discussion period is approaching. We have provided detailed responses and supplemented more experimental results. We hope they can address your concerns. Thanks for your valuable comments for improving the quality of our study. If you have other concerns, please do not hesitate to contact us, we are glad to resolve them.
>
> Kind regards,
>
> Submission 248 Authors

---

### Official Review · Reviewer_ETXB · 2023-07-24
**Comments for Submission248**

**Rating:** 4
**Confidence:** 4
**Correctness:** Yes
**Clarity:** Yes

**Strengths:**

The analyses of various adopted basic prompts in Fig.3 is interesting.

**Additional Feedback:**

See my comments above.

**Documentation:**

The github link is provided.

**Limitations:**

The authors well discuss the limitations in this work.

**Opportunities For Improvement:**

The authors need to provide some extra manual GTs to assist in evaluating the quality of pseudo GTs from SAM. Additionally, in terms of NeurIPS Benchmark Track, some benchmarking results are necessary, e.g., [1-5],

[1] ResUNet-a: A deep learning framework for semantic segmentation of remotely sensed data, 2020.
[2] Transformer-induced graph reasoning for multimodal semantic segmentation in remote sensing, 2022.
[3] Swin transformer embedding UNet for remote sensing image semantic segmentation, 2022.
[4] A novel transformer based semantic segmentation scheme for fine-resolution remote sensing images, 2022.
[5] LANet: Local attention embedding to improve the semantic segmentation of remote sensing images, 2020.

**Relation To Prior Work:**

Yes

**Summary And Contributions:**

This paper engages the SAM to enrich existing Remote Sensing data by providing extra object category, location, and instance information. This is useful for semantic segmentation, instance segmentation, and object detection. However, these pseudo annotations are highly relied on the robustness of pre-trained SAM. In other words, the author only uses SAM with some existing box prompts to obtain pseudo GTs, which is not enough for NeurIPS Benchmark Track.

---

> ### Author Response · Authors · 2023-08-26
> **# Response to Reviewer ETXB (Q1-Q2)**
>
> We sincerely thank you for the careful and thoughtful comments. Below we address the key concerns.
>
>
> __Q1: The authors need to provide some extra manual GTs to assist in evaluating the quality of pseudo GTs from SAM.__
>
> __A1__: Thanks for your comment. Indeed, it is unnecessary to manually annotate GT labels. In the HRSC2016 detection dataset, 124 images of the official testing set have ground truth segmentation labels annotated by experts. We use these labels to evaluate the quality of pseudo GTs from SAM, see Section 2.2:
>
> *...**Additionally, in the testing set, 124 images possess bounding box annotations and pixel-level labels simultaneously, making it highly suitable for evaluating the accuracy of SAM annotations. Therefore, we conduct an ablation study on the testing set consisting of the aforementioned 124 images to determine the optimal configuration for SAM.** Following this, we generate segmentation labels for the remaining datasets...*
>
>
> With these labels, we test different prompt settings and select the optimal setting for generating segmentation annotations according to the results in Table 1.
>
> __Q2: Additionally, in terms of NeurIPS Benchmark Track, some benchmarking results are necessary, e.g., [1-5] [1] ResUNet-a: A deep learning framework for semantic segmentation of remotely sensed data, 2020. [2] Transformer-induced graph reasoning for multimodal semantic segmentation in remote sensing, 2022. [3] Swin transformer embedding UNet for remote sensing image semantic segmentation, 2022. [4] A novel transformer based semantic segmentation scheme for fine-resolution remote sensing images, 2022. [5] LANet: Local attention embedding to improve the semantic segmentation of remote sensing images, 2020.__
>
>
> __A2__: Thanks for your suggestion. Since [2] has been retracted, we will not review it. In the revised manuscript, we compare our models to the other suggested methods on the ISPRS Potsdam dataset, see Table 3. We further evaluate the SEP on the IsAID dataset, and the comparison to existing methods is presented in Table 4.
>
> Here we only show related parts of these tables. Please refer to them in the revised manuscript for a complete comparison.
>
> *Table 3. Segmentation results of different methods on the ISPRS Potsdam dataset (Comparison method).*
>
> | Method | Pretrain | Backbone | OA | mF1 |
> | :------ | :-- | :--- | :---: | :---: |
> | ST-UNet [15] | --- | ResNet-50 | --- | 86.13 |
> | ResUNet-a d7v2 [9] | --- | --- |  91.50 | 92.90 |
> | LANet [11] | IMP | ResNet-50 |  90.84 | 91.95 |
> | DCFAM [39] | IMP | Swin-S |  92.00 | 93.25 |
>
> *Table 4. Segmentation results of different methods on the IsAID dataset. $\ddagger$: MAE pre-training on the MillionAID.*
>
> | Method | Pretrain | Backbone  | mIOU |
> | :------ | :-- | :--- | :---: |
> | HMANet [28] | IMP | ResNet-50  | 62.64 |
> | UperNet | MAE $\ddagger$ | ViTAE-B + RVSA$^\Diamond$ | 64.49 |
> | FactSeg [25] | IMP | ResNet-50 | 64.79 |
> | RssFormer [44] | IMP | RSS-B |  65.88 |

---

> ### Author Response · Authors · 2023-08-29
> **Request for Discussion**
>
> Dear Reviewer ETXB
>
> We sincerely thank you again for your great efforts in reviewing this paper. We have addressed your major concerns about the label quality and the comparisons with other methods. Please don't hesitate to let us know if there are still concerns/questions. Your insights are important and valuable to us.
>
> Kind regards,
>
> Submission 248 Authors

---

> > ### Comment · Reviewer_ETXB · 2023-08-31
> > **Thank authors for your efforts**
> >
> > Thank you for your efforts in addressing my questions. But my concerns are still remain. My main concern is that using advanced SAM to generate GT, i.e., generating a large-scale RS segmentation mask to an existing RS object detection dataset by SAM, is hard to be a benchmark for related researchers. The readers can't ensure the pseudo segmentation can be used in model training or evaluating, because the predicted masks by SAM is still far away than human annotations.
> >
> > In addition, the results of Table 1 does not show the reasonability of the proposed pipeline, because the requirement of segmentation accuracy of a RS segmentation dataset should be very high. However, the imperfect mIoU score of the proposed method indicates the predicated masks still have large errors and are not convincing as a benchmark segmentation dataset, to train and evaluate models.
> >
> > Therefore, I still remain my score. Because adapting SAM to generate predicted masks as Pseudo GT in an existing dataset (unsatisfied GT accuracy), is hard to be a benchmark for related community.

---

> > > ### Author Response · Authors · 2023-08-31
> > > **Response to Reviewer ETXB**
> > >
> > > We sincerely appreciate your efforts in reviewing our paper and providing valuable comments.
> > >
> > > Frankly speaking, we disagree with your opinion that the proposed SAMRS dataset cannot be used by other researchers for training and evaluation. We have carried out a series of pre-training experiments using this dataset to validate its effectiveness, as shown in Table 3-4 of the main paper. In addition, Table S3 in the supplementary material indicates that the evaluation results on the SAMRS validation set can reflect the model performance on fine-tuning. More importantly, besides this dataset, we develop a pipeline for efficiently generating RS segmentation annotations. Other researchers can use this pipeline to obtain a larger-scale segmentation dataset, which is useful in pre-training larger models.
> > >
> > > Again, thanks for your efforts in reviewing this paper and helping us improve its quality. We hope the above clarification can address your concern.

---

> > > > ### Comment · Reviewer_ETXB · 2023-08-31
> > > >
> > > > Thanks for the response. I have seen the Tables 3 and 4. I understand that the SAMRS can be as an extra large-scale source to pre-train models to improve performance. But this does not mean the SAMRS can be used as a benchmark. Because a basic semi-supervised method like Naive-student [1] can also achieve this objective, benefiting from the introduction of additional information within an existing RS dataset.  On the other hand, the authors did not address my concern about how to use the SAMRS as evaluation set. According to Table 1's mIoU score, it is obvious that the readers can not use the pseudo GTs in SAMRS dataset to evaluate the performance of related models. Therefore, I keep my score.
> > > >
> > > > [1] Naive-Student: Leveraging Semi-Supervised Learning in Video Sequences for Urban Scene Segmentation. ECCV, 2020.

---

> ### Author Response · Authors · 2023-08-31
> **Thanks for your reviews**
>
> Dear Reviewer ETXB
>
> The discussion period is approaching. We have provided detailed responses and supplemented more experimental results. We hope they can address your concerns. Thanks for your valuable comments for improving the quality of our study. If you have other concerns, please do not hesitate to contact us, we are glad to resolve them.
>
> Kind regards,
>
> Submission 248 Authors

---

### Author Response · Authors · 2023-08-26
**To all reviewers**

**Thanks to all reviewers for their valuable feedback!**

The revised main paper and supplementary material have been uploaded. We sincerely thank the reviewers for their thoughtful reviews. They were all taken into account to improve the quality of the manuscript, and we hope you will find the current version of the paper in accordance with the requirements. Please see our detailed point-by-point replies below. Citations from manuscript are marked with *italics* (except table), while the key texts are emphasized by **bold**.

We provide detailed responses to each reviewer respectively and promise will incorporate all feedbacks in the revised version.

---

### Decision · Program_Chairs · 2023-09-22

**Decision:**

Accept (Poster)

**Comment:**

This paper underwent a thorough review process by four reviewers. The reviewers were divided, with half recommending rejection and the remaining reviewers expressing support for acceptance. A primary concern raised by Reviewer ETXB is the suitability of SAMRS for evaluation purposes. After a thorough examination of the submission, it is evident that the primary intent behind SAMRS is to provide a large-scale dataset for pre-training, with less emphasis on positioning it as a benchmark dataset. The authors have not strongly highlighted SAMRS as a benchmark in their submission. I believe that SAMRS holds significant value as a resource for large-scale pre-training in the field RS images. Therefore, I recommend accepting this paper, provided that the authors include necessary discussions and incorporate new experiments into the final version.